# Nonlinear photovoltaic effects in monolayer semiconductor and layered magnetic material hetero-interface with *P*- and *T*-symmetry broken system

Shuichi Asada[1], Keisuke Shinokita [1], Kenji Watanabe [2], Takashi Taniguchi [3] & Kazunari Matsuda [1] ✉

Stacking two non-polar materials with different inversion- and rotational-symmetries shows unique nonlinear photovoltaic properties, with potential applications such as in next generation solar-cells. These nonlinear photo-current properties could be further extended with broken time reversal symmetry present in magnetic materials, however, the combination of time reversal and rotation symmetry breaking has not been fully explored. Herein, we investigate the nonlinear photovoltaic responses in van der Waals hetero-structure compromising of monolayer semiconductor and layered magnetic material, $MoS_2/CrPS_4$; a system with broken *P*- and *T*-symmetry. We clearly observe the finite spontaneous photocurrent as shift current at the interface of the $MoS_2/CrPS_4$ heterostructure. Moreover, we demonstrate that the spontaneous photocurrent drastically changes according to the magnetic phases of $CrPS_4$. The magnetic phase dependent spontaneous nonlinear photocurrent provides a platform for studying nonlinear photoresponses in systems with broken *P*- and *T*-symmetry, and the potential development of magnetic controllable photovoltaic devices.

Recently, the introduction of shift current concepts from topological physics drastically extend the possibilities for photovoltaic devices because of their unconventional spontaneous photocurrent properties[1–4]. The shift current as one of the bulk photovoltaic effects are the most promising photovoltaic responses to be able to overcome the Shockley–Queisser limit in the conventional p-n junction photo-voltaic devices[5–8]. These anomalous photovoltaic responses including extremely large open-circuit voltage, not limited by band-gap energy, and robust photocurrent to defects and impurities of materials, have been observed in the polar bulk materials[9–13]. The shift current could be explained as topological photocurrent depending on Berry con-nection, which is a vector in real space indicating the information of

center-of gravity position of electrons[14,15]. In the crystal with broken spatial inversion (*P*-) symmetry, the imbalance electron "shift" due to optical interband transitions before and after photoexcitation is occurred, which then generates the spontaneous photocurrent under zero-bias conditions as shift current. In addition to the shift current, the nonlinear topological photoresponses also induce the injection current, which is caused by difference of group velocity and imbalance photoexcitation with circular polarized light in *k*-space[16–18]. Since the shift current and injection current are induced by the linear- and circular-photogalvanic effects (LPGE and CPGE) of linearly and circu-larly polarized light, respectively, their effects are generally observed separately. However, in time-reversal (*T*-) symmetry broken crystals

[1]Institute of Advanced Energy, Kyoto University, Uji, Kyoto 611-0011, Japan. [2]Research Center for Functional Materials, National Institute for Materials Science, 1-1 Namiki, Tsukuba, Ibaraki 305-0044, Japan. [3]International Center for Materials Nanoarchitectonics, National Institute for Materials Science, 1-1 Namiki, Tsukuba, Ibaraki 305-0044, Japan. ✉e-mail: matsuda@iae.kyoto-u.ac.jp

such as some layered magnetic materials, each spin band is lifted by spin polarization and shows an imbalanced band structure in momentum space. The electron is photoexcited unevenly in terms of momentum, and the group velocity of the electron is not perfectly canceled, which would induce a linear injection current in systems with broken $T$-symmetry[19–22]. The shift current is also induced by the CPGE after the magnetic transition. These magnetic PGEs have odd parity under magnetization changes; thus, they are not observed in non-magnetic materials where $T$-symmetry is preserved[23].

The emerging two-dimensional (2D) materials, including semiconducting transition metal dichalcogenides have opened the new research fields in the fundamental science and potential applications in various electronic and optical applications[24–26]. The artificial van der Waals (vdW) heterostructures fabricated by stacking 2D materials lead to induce the emerging periodicity of crystal structures such as moiré superlattice[27–32] and provide us the new pathways to control the $P$-symmetry at their heterointerfaces. Recently, it has been reported the nonlinear photocurrent arising from shift current in the artificially $P$-symmetry breaking vdW heterointerface by stacking three-fold symmetry MoS$_2$ and two-fold symmetry black phosphrous[33]. Moreover, the artificial symmetry breaking would extend the exploration for high efficiency shift current photovoltaic devices[34–36], however, no promising candidates have been shown[37]. Accordingly, it is crucial to experimentally investigate the shift current and injection current depending on Berry connection and Berry curvature in the novel interface of vdW heterostructure with the systems with broken $P$- and $T$-symmetry. The understanding of nonlinear photocurrent responses in the vdW heterostructure with the system with broken $P$- and $T$-symmetry would provide the ground breaking importance for fundamental physics and applications for emerging photovoltaics.

In this study, we demonstrated the nonlinear photoresponses in monolayer semiconductor and layered magnetic materials vdW heterostructure with the system with broken $P$- and $T$-symmetry. The shift current along the direction of parallel to $P$-symmetry broken axis at the interface of MoS$_2$/CrPS$_4$ vdW heterostructure has been experimentally observed above 40 K. Moreover, we demonstrate the switchable spontaneous photocurrent arising from the magnetic phase transition of CrPS$_4$ in vdW heterostructure depending on the temperature and external magnetic fields. The detail physical mechanism of anomalous spontaneous photocurrents in the vdW heterostructure with the system with broken $P$- and $T$-symmetry will be discussed.

## Results

Figure 1a shows a schematic of photocurrent measurement of monolayer (1L)-MoS$_2$ and multi-layers CrPS$_4$ vdW heterostructure device. The rotational and $P$-symmetry by stacking of three-fold symmetry MoS$_2$ and two-fold symmetry CrPS$_4$ are diminished, and reduced at their interface[38–40], and only one-mirror plane is maintained at the vdW interface, as shown in the inset of Fig. 1a. The $(x, y, z)$ axes are defined for the crystal structure as shown in the inset of Fig. 1a. Figure 1b shows the optical image of monolayer MoS$_2$ and multi-layers CrPS$_4$ vdW heterostructure device. The monolayer MoS$_2$ and multi-layers CrPS$_4$ were mechanically exfoliated from bulk single crystals, respectively, and then the monolayer MoS$_2$ was stacked on multi-layers CrPS$_4$ by dry transfer method (see the "Methods" section). The twist-angle between MoS$_2$ and CrPS$_4$ was controlled to match each mirror planes with judging from each characteristic cleaved edge angle of 67.5° and 60° for multi-layers CrPS$_4$ and monolayer MoS$_2$, respectively[41–43]. The heterostructure fabrication was confirmed from photoluminescence spectrum as shown in Fig. 1c.

Figure 1d shows the current–voltage ($I$–$V$) characteristics measured along the $x$-axis of MoS$_2$/CrPS$_4$ vdW heterostructure device, under dark (black) and illumination of linearly polarized light of 532 nm (red) at room temperature. The $I$–$V$ characteristics under dark conditions show simple linear behavior in the current when the applied bias voltage is not zero. The significant finite value of spontaneous

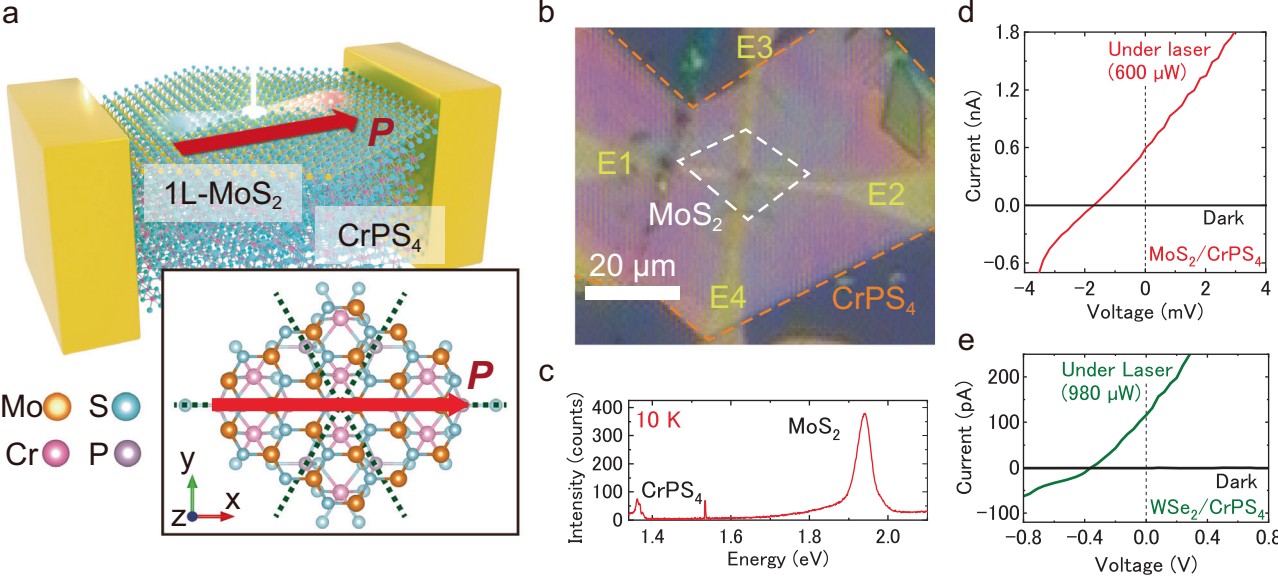

**Fig. 1 | Schematic of 1L-transition metal dichalcocogenide/CrPS$_4$ vdW heterostructure device. a** Schematic of photocurrent measurement of 1L-MoS$_2$/CrPS$_4$ vdW heterostructure device. The inset shows schematic of in-plane crystal structure of 1L-MoS$_2$/CrPS$_4$ vdW heterostructure. The green and red dotted lines show the axis of inversion symmetry MoS$_2$ and CrPS$_4$, respectively. In the heterointerface, the inversion symmetry is reduced to maintain in only one mirror plane and rotational symmetry is canceled. Consequently, the spontaneous polarization indicated by the red arrows with broken $P$-symmetry are induced, according to the overlapped mirror plane. **b** Optical image of 1L-MoS$_2$/CrPS$_4$ vdW

heterostructure device. The scale bar of 20 μm is shown in the images. The 1L-MoS$_2$ was stacked on multi-layer CrPS$_4$, and four-electrodes were fabricated along to parallel and perpendicular to expected polarization direction.
**c** Photoluminescence (PL) spectrum of 1L-MoS$_2$/CrPS$_4$ vdW heterostructure. The PL peaks at 1.92, and 1.38 eV correspond to the emissions from 1L-MoS$_2$, and CrPS$_4$, respectively. **d** $I$–$V$ curve of 1L-MoS$_2$/CrPS$_4$ vdW heterostructure device under dark and laser illumination of 532 nm conditions. **e** $I$–$V$ curve of 1L-WSe$_2$/CrPS$_4$ vdW heterostructure device under dark and laser illumination of 532 nm conditions.

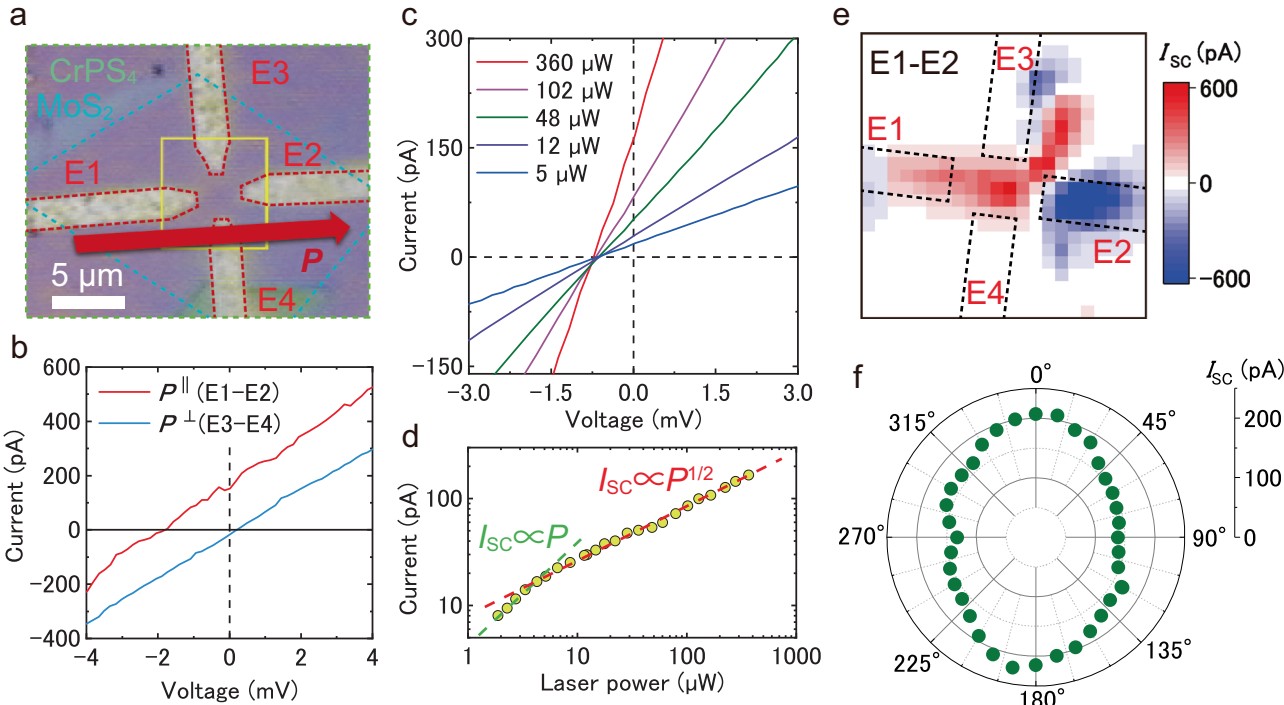

**Fig. 2 | Photovoltaic properties of 1L-MoS$_2$/CrPS$_4$ vdW heterostructure device in room temperature. a** Optical image of 1L-MoS$_2$/CrPS$_4$ vdW heterostructure with four electrodes configurations. The electrodes are highlighted in red dotted lines, and each electrode is assigned as E1–E4 to clarify measurement direction. The scale bar of 5 µm is shown in the images. Hereinafter, measurement results without annotations indicate the results between the E1 and E2 along to predicted polarization direction. **b** I–V characteristics of 1L-MoS$_2$/CrPS$_4$ vdW heterostructure under the laser illumination of 532 nm, which were measured by different current detection configurations with parallel (E1 and E2) and perpendicular (E3-E4) to expected polarization. **c** I–V characteristics of 1L-MoS$_2$/CrPS$_4$ vdW heterostructure with various excitation powers. **d** Excitation laser power dependence of spontaneous photocurrent of 1L-MoS$_2$/CrPS$_4$ vdW heterostructure. In the low power region, the spontaneous photocurrent is proportional to the laser power, whereas the square-root dependence of the photocurrent is observed in the high power region. **e** Photocurrent mapping measured between the E1 and E2 electrodes, indicated by the dotted lines. **f** Polar plot of spontaneous photocurrent with linear polarized light along to E1 and E2 direction. 0° and 180° correspond to E1 and E2 electrodes, respectively.

photocurrent at zero-bias voltage is clearly observed under laser illumination, which is in contrast to no spontaneous photocurrent, i.e. the zero-current at zero-bias voltage under dark condition. Moreover, the spontaneous photocurrent under light illumination is also observed in the WSe$_2$/CrPS$_4$ vdW heterostructure based on different monolayer semiconductor, as shown in Fig. 1e (see details in Fig. S1). Spontaneous photovoltaic effects were not observed in the devices with only the MoS$_2$ monolayer or bulk CrPS$_4$ (see Figs. S2 and S3). These results strongly imply that the experimentally observed spontaneous photocurrent is a universal behavior in the monolayer semiconductor transition metal dichalcocodenide/CrPS$_4$ vdW heterostructure device.

Figure 2a shows the optical image of MoS$_2$/CrPS$_4$ vdW heterostructure device with four electrodes (E1–E4) to investigate the direction of nonlinear spontaneous photocurrent, where the four electrodes indicated by red dashed lines are fabricated in the center of MoS$_2$/CrPS$_4$ heterostructure region. The photocurrent measurements using E1 and E2 (E3 and E4) correspond to photocurrent generation along the x- and y-axes, respectively. It has been well known that the shift current strongly depends on spontaneous polarization direction[33,44]. The significant spontaneous photocurrent at zero-bias voltage is clearly observed along to the x-axis and mirror plane, while no spontaneous photocurrent is observed along to the y-axis even under the same photoexcitation conditions, as shown in Fig. 2b.

Figure 2c shows the I–V characteristics with various laser intensities from 2 to 360 µW measured between E1 and E2 electrodes. The spontaneous photocurrent at zero-bias voltage increases with increasing the laser intensities. The laser power dependence of spontaneous photocurrent is plotted in Fig. 2d. The spontaneous photocurrent linearly increases as a function of laser power in the weak

power conditions below 5 µW, while the spontaneous photocurrent gradually saturates in the high power conditions. The behavior of nonlinear power dependence of spontaneous photocurrent is much different from the linear power dependence from photovoltaic responses of p–n junction and Schottky barrier[45–47]. The possible mechanism for the nonlinear spontaneous photocurrent is one of the second-order nonlinear optical responses, the shift current, which is caused in the system with broken P-symmetry arising from the shift of center of gravity electron position in real space before and after photoexcitation[14]. The nonlinear shift current $J_{shift}$ as a function of excitation power P is described as follows:

$$J_{shift} = S_0 \cdot \frac{P}{\sqrt{S_1 \cdot P + S_2}} \tag{1}$$

where $S_0$, $S_1$, and $S_2$ are coefficients, which contribute to the shift current[4,48]. When P is enough small, $S_2$ becomes dominant compare with $S_1$, and $J_{shift}$ shows linear dependence as a function of P, as shown in the green dotted line in Fig. 2d. Moreover, $S_1$ becomes dominant with increasing P and shows $P^{1/2}$ power dependence, as shown in the red dotted line in Fig. 2d. The experimental result of spontaneous photocurrent as a function of excitation power is well reproduced by the calculated result using Eq. (1). This nonlinear photocurrent is much contradict to the linear photocurrent arising from Schottky barrier[49], which implies that the experimentally observed spontaneous photocurrent comes from the shift current.

Figure 2e shows the spontaneous photocurrent mapping in MoS$_2$/CrPS$_4$ vdW heterostructure between E1 and E2 under laser light illumination of 532 nm and an excitation power of 600 µW. The positive

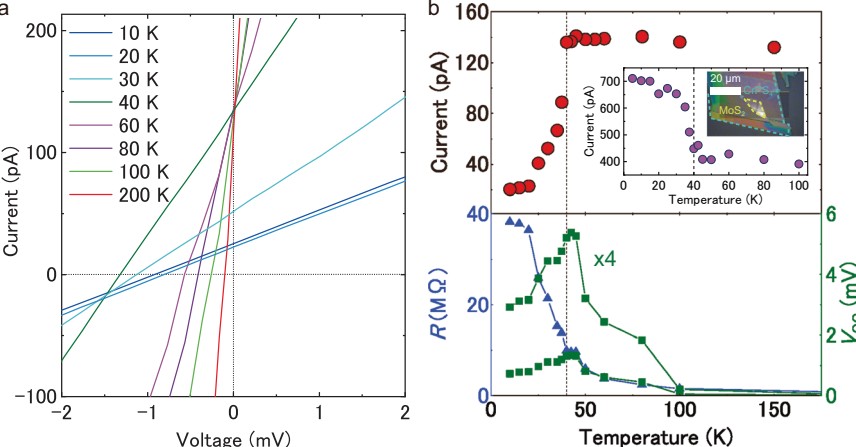

**Fig. 3 | Temperature dependence of photovoltaic properties of MoS₂/CrPS₄ vdW heterostructure device. a** Temperature dependence of $I$–$V$ characteristics of MoS₂/CrPS₄ vdW heterostructure under the light illumination from 10 to 200 K. **b** Spontaneous photocurrent at zero-bias voltage as a function of temperature in the upper panel. The inset shows the temperature dependence of spontaneous photocurrent, and optical image in another MoS₂/CrPS₄ vdW heterostructure device. The open-circuit voltage, as indicated by green dots and lines in the lower panel and effective series resistance calculated by the gradient of $I$–$V$ curve, as indicated by the blue dots and line in lower panel. The black dotted line at 40 K indicates Neel temperature of CrPS₄.

and negative photocurrent signals around each electrode due to Schottky barrier internal voltage between heterostructure and metal electrodes are clearly observed in the photocurrent mapping. More importantly, the large spontaneous photocurrent is observed in the center of heterostructure region far from electrodes (see Fig. S4), and the spontaneous photocurrent in the center strongly depends on the measurement direction (see Fig. S5). These results strongly imply that the experimentally observed spontaneous photocurrent in the heterostructure region comes from intrinsic photovoltaic properties in the monolayer semiconductor/CrPS₄ heterointerface.

Figure 2f shows the laser polarization angle dependence of spontaneous photocurrent in MoS₂/CrPS₄ vdW heterostructure (also see in Fig. S6), in which the linearly polarization angle of incident laser light is defined as relative angle from the mirror-plane ($x$-axis) of heterointerface. The polar plot of spontaneous photocurrent as a function of polarization angle shows weakly anisotropic response with two-fold symmetry, along with symmetric axis of parallel to mirror-plane in MoS₂/CrPS₄ vdW heterointerface. According to the theoretical calculation, the polar plot of the spontaneous bulk photovoltaic effect with only monolayer MoS₂ with $C_3$ symmetry is expected to show threefold symmetry and both positive and negative signals[33,50]; however, the polar plot of the spontaneous photocurrent of the MoS₂/CrPS₄ vdW heterostructure shows twofold symmetry along the spontaneous polarization direction and consistently positive signals, as shown in Fig. 2f. Thus, the experimentally observed spontaneous photocurrent is caused by the shift current induced by the broken $P$-symmetry at the MoS₂/CrPS₄ vdW heterointerface. However, these results are characteristic of bulk photovoltaics, which may not occur due to the shift current alone. In addition to the shift current, the ballistic current might also be considered a bulk photovoltaic effect that occurs under linearly polarized light excitation[51,52]. However, the major difference between the shift current and ballistic current is the response to circularly polarized light[53,54]. As shown in Fig. S7, the spontaneous photocurrent in the MoS₂/CrPS₄ heterostructure device disappears under circularly polarized light (CPL), suggesting that the effect of the ballistic current is negligible. The measured spontaneous current under CPL in Fig. S7 comes from a nonmagnetic injection current, for example, from a ballistic current induced by CPL. The experimentally observed injection current is a novel result that has not been reported for artificial vdW heterointerfaces, which provides new insights into the bulk photovoltaic effect at artificial vdW heterointerfaces.

The nonlinear spontaneous photocurrent induced by the breaking of $T$-symmetry in addition to $P$-symmetry is investigated in the MoS₂/CrPS₄ vdW heterostructure, because the vdW heterointerface is formed between the monolayer semiconductor and magnetic material of CrPS₄. Figure 3a shows the temperature dependence of $I$–$V$ curves in the MoS₂/CrPS₄ vdW heterostructure under the illumination of laser with a wavelength of 532 nm and power of 600 μW. With decreasing temperature from 300 to 40 K, the gradient of $I$–$V$ curve decreases and open-circuit voltage increases. The fill factor is -0.25 independent of temperature, as shown in Fig. S8, due to the linear $I$–$V$ characteristics of the bulk photovoltaic effect. The effective series resistances in the vdW heterostructure are evaluated from the gradient of the linear $I$–$V$ curves. The lower panel of Fig. 3b shows the effective series resistance (blue dot and line) and open-circuit voltage (green dot and line) of MoS₂/CrPS₄ vdW heterostructure as a function of temperature. The effective series resistance increases with decreasing temperature until 40 K, which corresponds to the temperature dependence of resistivity in monolayer MoS₂[55,56]. Moreover, the open-circuit voltage also increases with decreasing temperature above 40 K.

Figure 3a shows the drastically change of $I$–$V$ curve with decreasing temperature below 40 K. The spontaneous photocurrent shows almost constant values above 40 K as shown in Fig. 3a, while the most significant change in the temperature dependence of $I$–$V$ curve is drastically decrease of spontaneous photocurrent below ~40 K. Figure 3b shows the plot of spontaneous photocurrent as a function of temperature. The critical decrease of spontaneous photocurrent is celery observed below 40 K. Moreover, the trend of open-circuit voltage also changes below the critical temperature of 40 K, as shown in the lower panel of Fig. 3b.

The critical temperature of 40 K well corresponds to the magnetic transition temperature from paramagnetic to anti-ferromagnetic (AFM) phase of CrPS₄ as Neel temperature ($T_N$ = 38 K)[57–60]. It is expected that the shift current as physical origin of spontaneous photocurrent shows temperature independent behavior, because it comes from the center of gravity position shift of excited electrons in the photoexcitation process, which is not to be affected by the defect and atomic registry[61,62]. However, the spontaneous photocurrent significantly decreases below Neel temperature of 40 K in CrPS₄, which strongly implies the physical mechanism of nonlinear spontaneous photocurrent in addition to shift current in the system with broken $T$-symmetry and $P$-symmetry of MoS₂/CrPS₄ heterointerface at low temperature.

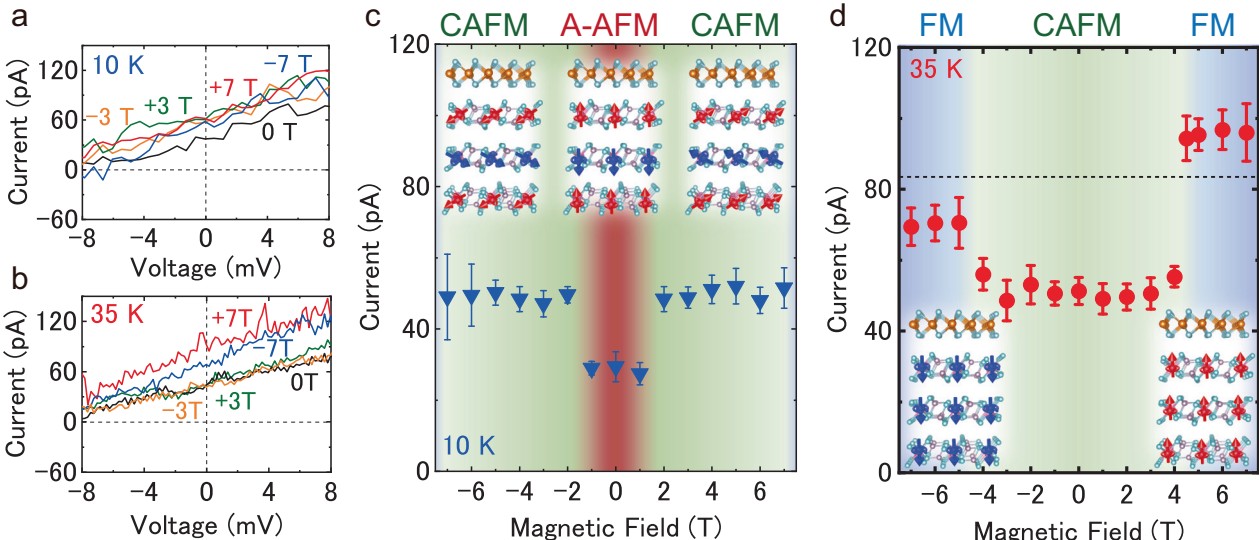

**Fig. 4 | External magnetic field dependence of photovoltaic properties of MoS₂/CrPS₄ vdW heterostructure device. a** and **b** *I–V* characteristics of MoS₂/CrPS₄ vdW heterostructure with external out of plane magnetic field from −7 to 7 T at 10 K (**a**) and 35 K (**b**). **c** and **d** External magnetic field dependence of spontaneous photocurrent is represented by red dots at 10 K (**c**) and 35 K (**d**). Each colored region shows the magnetic state of CrPS₄, ferromagnetic (FM, blue), canted anti-ferromagnetic (CAFM, green) and A-type anti-ferromagnetic (A-AFM, red), respectively[60]. The black dotted line in **d** shows the value of shift current above 40 K. The schematic of crystal and spin structures of 1L-MoS₂/CrPS₄ vdW heterostructure are shown in the figures.

We will discuss the anomalous and critical change of spontaneous photocurrent below 40 K. The shift current is described as the product of terms of optical transition probability and shift vector[14]. Noted that the optical transition probability measured by the differential reflectance spectra in the vdW heterostructure does not change below 40 K of Neel temperature in the multi-layers CrPS₄ (see in Fig. S9), which suggests that the factor of optical transition probability is not the main physical reason of the critical change of spontaneous photocurrent. The magnetic phase change of CrPS₄ at the heterointerface would affect the shift vector, however, it has revealed that the shift vector itself does not depend on the *T*-symmetry[63]. The magnetic lattice distortion accompanied by magnetic phase transition would also cause the possibility of change of shift-vector[64], however, we could not observe any clear changes of Raman spectrum below 40 K as shown in Fig. S10. Moreover, the temperature dependence of lattice constant in bulk CrPS₄, especially, *x*-axis along to expected polarization direction shows continuously change from 300 to 10 K straddling 40 K according to the previous result[39] (see Fig. S11). Noted that the change of lattice constant is small value of only about 0.03% between 40 and 10 K. These are much contradict to the experimental results of constant values of spontaneous photocurrent from 300 to 40 K and critical change below 40 K, as shown Fig. 3a. Moreover, the spontaneous photocurrent also shows different behavior of clear enhancement below Neel temperature in another device of MoS₂/CrPS₄ vdW heterostructure, as shown in in the inset of Figs. 3b and S12. These experimental results strongly suggest that the change of optical transition probability and shift vector induced by magnetic lattice distortion cannot explain the reversible change of shift current depending on the device. Thus, the experimentally observed significant and critical change of spontaneous photocurrent below 40 K does not come from the change of shift current.

As described above, the shift current does not be affected by *T*-symmetry of the system, although, the experimental results of spontaneous photocurrent in MoS₂/CrPS₄ vdW heterostructure strongly depend on the magnetic state of CrPS₄ layer, which implies additional nonlinear photocurrent mechanism in the system with broken *P*- and *T*-symmetry. The recent theoretical studies of nonlinear photocurrent predict the generation of magnetic injection current due to imbalance photoexcitation of electrons in momentum space in layered magnetic materials such as CrPS₄ with the system with broken *P*- and *T*-symmetry[19,22,23]. The spontaneous photocurrent in AFM CrPS₄ below Neel temperature ($T_N \sim 40$ K) might be affected by the generation of magnetic injection current. As shown in Fig. S13, bulk CrPS₄ does not have a spontaneous photocurrent even below the Neel temperature, which suggests that the magnetic injection current is induced by a system with broken *P*-symmetry, such as the heterointerface of MoS₂ and CrPS₄. The experimental result of the critical decrease in the spontaneous photocurrent shown in Fig. 3a (also shown in Fig. S12) comes from the generation of a magnetic injection current with a negative sign superimposed on the temperature-independent shift current, which will be discussed later. The generation of a magnetic injection current and shift current due to a decrease in temperature and a change in the magnetic order state are shown in Table S1.

In order to experimentally investigate the relationship between spontaneous photocurrent and magnetic states of CrPS₄, we measured external magnetic field dependence of photocurrent in the vdW heterointerface below 40 K. Figure 4a, and b show the *I–V* curves in MoS₂/CrPS₄ vdW heterostructure with applying out of plane magnetic fields at 10 and 35 K below Neel temperature. The relationship between the magnetic state of CrPS₄ at each temperature and the out-of-plane magnetic field is shown in Fig. S14. The *I–V* curve changes and the spontaneous photocurrent increases with the sweep of out of plane magnetic field from 0 to 3 T (from 0 to −3 T) at 10 K, as shown in Fig. 4a. Figure 4c shows the spontaneous photocurrent as a function of out of plane magnetic field at 10 K. Figure 4c and d show the results under the sweeping of magnetic field from 0 T. The effect of hysteresis on the spontaneous photocurrent is shown in Fig. S15. The spontaneous photocurrent quickly increases at the boundaries of ±2 T, which is caused by the magnetic field induced phase transition from A-type AFM with out-of-plane ferromagnetic ordered spins to canted-antiferromagnetic state with ordered canted spins within a layer (A-AFM) and canted anti-ferromagnetic ordered spins (CAFM) of CrPS₄[60]. The experimental result of increase of spontaneous photocurrent in Fig. 4c comes from decrease of negative sign magnetic injection current depending on the magnetic phase of CrPS₄ superimposed to shift current triggered by phase transition from A-AFM to CAFM of CrPS₄ by external out of plane

magnetic field in the vdW heterostructure, because the value of magnetic injection current in A-AFM state is larger than that in CAFM state. As shown in Fig. S16, electrical resistance $R$ is constant with respect to the external magnetic field; thus, the change in spontaneous photocurrent clearly does not result from the change in carrier mobility.

The $I$–$V$ curve changes and the spontaneous photocurrent increases with the sweep of out of plane magnetic field from 0 to 7 T (from 0 to −7 T) at 35 K, as shown in Fig. 4b. Figure 4d shows the magnetic field dependence of spontaneous photocurrent at 35 K near Neel temperature. The spontaneous photocurrent increases with the sweep of out of plane magnetic field from 3 to 7 T (from −3 to −7 T) at 35 K. The spontaneous photocurrent also quickly increases at the boundaries of ±4 T, which is caused by the magnetic field induced phase transition from CAFM to ferromagnetic (FM) ordered state of CrPS$_4$. Above external magnetic field of ±4 T, the values of spontaneous photocurrent reach to almost same value without magnetic field above Neel temperature of 40 K, which is indicated by the dotted line in Fig. 4d. The stepwise increase of spontaneous photocurrent in Fig. 4d comes from the disappearance of magnetic injection current and residual shift current in the spontaneous photocurrent by magnetic phase transition from CAFM to FM of CrPS$_4$ by external magnetic field.

The sign of magnetic injection current is determined by the up- and down-spin of top layer in the AFM state, and reversed AFM[19,22,23]. The two-types of heterointerface of MoS$_2$/CrPS$_4$ vdW heterointerface are possible in AFM, and reversed-AFM of CrPS$_4$ below Neel temperature ($T_N$). The experimental results of critical increase, and decrease of spontaneous photocurrent induced by A-AFM transition, as shown in Figs. 3a, and S12 are well consistent with the cases of vdW heterointerface with AFM, and reversed-AFM of CrPS$_4$, respectively. The magnetic state switchable spontaneous photocurrent phenomena described above are understood by the change of magnetic injection current superimposed to shift current in vdW heterointerface. These results strongly imply the experimental observation of characteristic phenomena of nonlinear spontaneous photocurrent in vdW heterostructure with the system with broken $P$- and $T$- symmetry.

## Discussion

We studied nonlinear photocurrent responses of MoS$_2$/CrPS$_4$ vdW heterostructure with the system with broken $P$- and $T$-symmetry. The finite spontaneous photocurrent arising from shift current along the parallel axis to mirror plane at the interface of MoS$_2$/CrPS$_4$ has been observed. Moreover, we demonstrated that the spontaneous photocurrent drastically changes below Neel temperature of A-type AFM CrPS$_4$ in MoS$_2$/CrPS$_4$ heterostructure. The spontaneous photocurrent also changes depending on external magnetic field below the Neel temperature, accompanied by field induced magnetic phase transition. The critical suppression (enhancement) of spontaneous photocurrent below Neel temperature is caused by competition of the shift current and sign of magnetic injection current in MoS$_2$/CrPS$_4$ vdW heterointerface. Our results of magnetic switchable photovoltaic responses demonstrated here provide the new aspects on the nonlinear photovoltaic effects in $P$- and $T$-symmetry breaking vdW heterointerface, and new strategy for next-generation solar cells.

## Methods

### Sample preparation

Monolayer (1L) MoS$_2$ and thick multi-layers CrPS$_4$ were prepared on 270-nm-thick SiO$_2$ on Si substrates by mechanical exfoliation from their respective bulk single crystals. The thickness of MoS$_2$ was determined from the optical contrast and photoluminescence (PL) spectra. The MoS$_2$/CrPS$_4$ heterostructure was fabricated by a polydimethylsiloxane (PDMS)-based viscoelastic transfer technique using poly-methyl methacrylate (PMMA) stamp. The PMMA layer was washed away by immersing the sample in an acetone solution.

The microelectrode pattern was fabricated by photolithography method, and Bi/Au electrodes were thereafter fabricated by thermal evaporation method. The stacking angle $\theta$ defined by the relative angle between zigzag direction of 1L-MoSe$_2$ and $a$-axis of CrPS$_4$ in the heterostructure was evaluated to be $0 \pm 1°$ by the edge angle in optical image[38,39,65]. The correspondence between the characteristic edge angles and the crystal structure was confirmed by polarized Raman scattering and SHG measurements (Figs. S17 and S1)[8,66,67].

### Photocurrent measurement

A linearly polarized green laser (532 nm) was used as an excitation light source for the photocurrent measurements. The $I$–$V$ characteristics of the device was measured in the shield box (Keithley, 8101-PIV) at room temperature and in the cryostat (Janis Research, ST-500-UC) for the temperature variable measurements, respectively. The micro-Raman setup (Nanophoton, Ramantouch) was used for the measurement of PL, Raman scattering, and differential reflectance spectra. The source-meter (Keithley, 2636B) was used for applying the bias voltage and measure of current.

### Magnetic photocurrent measurements

A linearly polarized green laser (520 nm) was used as an excitation light source for $I$–$V$ characteristics measurement under out of plane magnetic field. The device was cooled by the He-flow cryostat (Janis Research, ST-500), and the magnetic field was generated by the superconductive magnet (Cryogenic, mCFM). The sourcemeter (Keithley, 2614B) was used for applying the bias voltage and measuring the current under magnetic fields.

## Data availability

Data presented in this paper and the supplementary materials are available from the corresponding author upon request.

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

## Acknowledgements

This work was supported by JSPS KAKENHI (Grant Nos. JP16H00910, JP16H06331, JP17H06786, JP19K14633, JP19K22142, JP20H05664, JP21H05232, JP21H05235, JP21H01012, JP21H05233, JP22K18986 and JP23KJ1381), JST FOREST program (Grant No. JPMJFR213K), JST CREST program (Grant No. JPMJCR24A5), the Collaboration Program of the Laboratory for Complex Energy Processes, Institute of Advanced Energy, Kyoto University. Growth of *h*-BN was supported from Grant Nos. JPMXP0112101001, JSPS KAKENHI and JP20H00354.

## Author contributions

S.A. contributed to the fabrication of samples studied in this work. S.A., K.S., and K.M. designed the experiments, which were performed by S.A. K.W. and T.T. provided *h*-BN crystal which used in samples. Data analysis was performed by S.A. and K.M. The draft was written by S.A., K.S., and K.M., with all authors contributing to reviewing and editing. The project was supervised by K.M.

## Competing interests

The authors declare no competing interests.
