## [Peer Review File · Nature Communications]

Nonlinear photovoltaic effects in monolayer semiconductor and layered magnetic material hetero-interface with P- and T-symmetry broken system

Corresponding Author: Professor Kazunari Matsuda

Version 0:

Reviewer comments:

Reviewer #1

(Remarks to the Author)

In this work, the authors report the experimental demonstration of a magnetically switchable linear photogalvanic effect (LPGE) in an interface material. The authors have performed all the standard checks to demonstrate that this is a PGE (linear dependence on power, polarization dependence, spot position dependence to eliminate contact effects), as well as clear and sharp dependence on the magnetization state. There has been great interest in the community to show the existence of this effect, and in my opinion this proof alone would be merit for publication.

However, I must also emphasize that the presentation, organization and motivation of the manuscript could use a very significant improvement. Below the authors can find a list of recommendations to ensure the paper can be accepted for publication and has broader impact in the community.

- The authors need to explain the difference between shift and injection currents more clearly in the introduction. The key difference is that in non-magnetic systems, shift current gives linear PGE, while injection gives only circular PGE. In magnetic systems injection provides an extra LPGE, which is what the authors are observing here. In addition the shift current might change too after the magnetic transition, but the linear injection current is odd under magnetization change, as the authors also observe.

- The authors should compare their results with Song, *Sci. Adv.* 7, eabg8094 (2021), where switchable photocurrents were reported in CrI₃. What is the difference between these works, and how does this work represent an improvement? The authors only provide two references on magnetic injection current which do not appear in the reference list (16-17). Presumably these are (18-19) which are repeated twice. I think it would be fair to cite the original works on shift current like Belinicher *Sov. Phys. Usp* 23, 199 (1980), Sipe *PRB* 61, 5337 (2000), as well as the recent ones on magnetic photocurrents Fei *PRB* 102, 035440 (2020), Watanabe *PRX* 11, 011001 (2021).

- The authors need to clarify their discussion about symmetry by using standard names for symmetry operations, and by using only one name for each symmetry. There is no such thing as an "axis of inversion symmetry". The inversion operation $(x,y,z) \rightarrow (-x,-y,-z)$ is done with respect to a point, called inversion center. When only one coordinate is inverted $(x,y,z) \rightarrow (-x,y,z)$ this is called a mirror plane. The authors later refer to the aligned vertical mirror planes of the heterostructure, which is the correct name. Inversion symmetry is always broken at every interface, but this does not necessarily generate a polar axis, as the authors seem to imply. To clarify the discussion, the authors should draw standard coordinate axis, define the symmetry operations with respect to that axis, and explain the symmetry constraints on the photocurrent. The sentences "parallel to the spatial inversion symmetry broken axis at the interface", "uni-axis of inversion", "inversion... is maintained only in the uniaxis" should be removed.

- The authors say they show dI/dV "along the expected polarization direction", what polarization is this? Can the authors please use coordinate axes to describe this? Is there any proof that there is charge polarization in the plane, and not out of plane or anywhere in between?

- Other use of language: Threefold-symmetry MoS₂ \rightarrow Threefold symmetric MoS₂. "Time reversal symmetry broken system" -

> System with broken time-reversal symmetry. Avoid repeating parenthesis in inversion (P-) and time reversal (T-) throughout the text. Avoid use of parenthesis as in "The increase (decrease) in photocurrent is caused by a decrease in the negative (positive) current", which make the text harder to read.

- The manuscript contains unnecessary repetition, for example in the description of what the shift current is which appears several times in the text.

- In my opinion, the authors use the word "topological" without need. It plays no role in the explanation and might as well be removed.

- The authors appear to use normal incident light but this is not stated anywhere in the text.

I believe these changes will lead to a significant improvement of the presentation of what is otherwise a very interesting and timely experiment.

Reviewer #2

(Remarks to the Author)

Photovoltaic effects in P- and T-symmetry broken systems were reported by Asada et al., which is intriguing and fundamentally promising in the field. At a specific temperature, the combined effect of shift and injected current is correlated with the observed PV effect. It thus demonstrates the need for the P- and T-broken system. However, there are a few important aspects that need to be addressed before publication.

1. The authors solely focused on the Shift current response when examining the overall photovoltaic effect. The bulk photovoltaic effect is a result of the combined response of ballistic and shift current. It's unclear why the Authors overlooked the ballistic contribution.

2. The light polarisation angle-dependent ISC does not verify materials exhibit shift current response. It is the characteristic of the bulk PV phenomenon. It is also applicable to the non-linear current versus intensity characteristics. It could be due to competition between the Ballistic and Shift current response. The authors can determine the relaxation time of the carriers in order to determine the contribution that solely originates from the Shift current response. The band alignment in the P and T-symmetry broken system can also shed light on the mechanism.

3. Materials show the above bandgap PV effect in accordance with the bulk PV phenomenon, which may allow them to surpass the efficiency S-Q limit. The observed VOC in the system is nowhere close to the above bandgap voltage. Please comment on this. From here, one might think that this could simply be a contribution to the interface. How do authors eliminate this contribution?

4. Increasing current response below 40 K could also be a consequence of decreasing resistance in the materials. From Fig. 3a, it is also clear that decreasing the temperature follows the decreasing ISC trend by compensating for increasing VOC. What about the trend in Fill Factor? What is the trend of R and VOC response in various magnetic fields?

5. It is crucial to provide the ferroelectric response at the interface for different temperatures.

6. Without MoS₂, CrPS₄ is a T-symmetry broken system. What is the behaviour of temperature-dependent injection current in only a T-symmetry broken system? Does it show injection current? Does it depend on the circular polarisation angle?

7. The magnitude of the current response in the inset of Fig. 3b (upper panel) is quite high. It is not clear how the experiment is performed. Is it measured with bias? It is mentioned that it is measured in different devices. How can similar device architecture display significantly different current values with opposite trends? Authors should address this behaviour.

Minor comments:

1. What is the intensity of the light used in PV experiments?

2. What would be expected PV outcomes when the magnetic field is parallel to the polarization direction? Do the authors expect any change in the overall response?

3. Experimental data in the Fig. 4a and b are not smooth. Also, the scale in the y-axis is plotted only in the positive value. It is important to show full-scale reading.

4. Page-5, "The I-V characteristics under dark conditions exhibit simple linear behavior...to the usual photoconductive effect". How I-V response in the dark can be related to the photoconductive effect?

5. VOC obtained in the magnetic PV measurements is nearly one order of magnitude higher than the normal I-V measurements presented in Fig. 3. Why are the measurements inconsistent?

Reviewer #3

(Remarks to the Author)

The manuscript "Nonlinear Photovoltaic Effects in Monolayer Semiconductor and Layered Magnetic Material Hetero-Interface with P- and T-Symmetry Broken System" by Shuichi Asada and colleagues investigated the photocurrent generated in a MoS₂/CrPS₄ heterostructure system with different magnetic states of CrPS₄. They use temperature and magnetic field as tuning knobs to access different magnetic states of CrPS₄ and, consequently, different photocurrents. Studying bulk photovoltaics in magnetic systems is certainly interesting and timely. Previously, much of the attention was confined to bulk photovoltaics in non-magnetic, non-centrosymmetric systems where the injection current can only be generated by circular polarized light and the shift current can only be generated by linear polarized light. By breaking time-reversal symmetry, the injection current can be generated by both linear and circular light through different mechanisms, and the shift current can also be generated by both linear and circular light through different mechanisms. This is an interesting area to explore both theoretically and experimentally. In the paper, the authors claimed the observation of magnetic injection due to linear polarized light, which is the main novelty of this work. However, the current evidence and presentation do not

clearly support this claim.

Major concerns:

1. As mentioned above, since the system simultaneously breaks P and T below the Néel temperature, linearly polarized light can generate both shift and magnetic injection currents. In other words, both effects are allowed to exist in the magnetic phase with linear light. It is not clear how the authors rule out the possibility of shift currents for the observed photocurrent. The observation in Fig. 4, showing the change in photocurrent with magnetic field, is not strong evidence of magnetic nature of the photocurrent in my opinion, as the resistance of the sample could change with different magnetic states, potentially inducing changes in photocurrents. Since magnetic injection current is the main novelty of this work, the authors need to provide strong evidence and explanations to substantiate this claim.
2. Related to the above question, one of the important pieces of evidence for magnetic injection should be its close relation with the magnetic structure. To me, the most important data to show the magnetic origin is the comparison between Fig. 3b and Fig. S7b, where the authors claim that when the spin configuration of the topmost layer of CrPS4 becomes opposite, the photocurrent also reverses. My concern is that this comparison is made between two different devices that were only conjectured to have opposite spin configuration without independent evidence. Why can't the authors change the spin configuration with an external B field? Relatedly, can the authors show forward and backward magnetic field scans, and is there a hysteresis? If the relation between the magnetic structure and photocurrent can be established experimentally, can the authors explain better how the top-layer spin of CrPS4 changes the injection current?
3. What is the temperature for the measurement presented in Figure 2? It seems to be in the non-magnetic phase above 40 K. In this non-magnetic regime, is there any difference of the claimed effect from a previous paper (Science 372, 68–72, 2021) which reported the observation of a large shift current in WSe2 + black phosphorus, where an in-plane polarization is created at the interface?
4. The description and analysis of the polarization dependence in Fig. 2, starting in line 160, are confusing. Particularly, how was Fig. 2f measured? Did the authors fix the current collection direction or not? If the current is only collected along a fixed direction while changing the polarization, then even for a system with out-of-plane C3 symmetry, the polarization dependence should be two-fold instead of three-fold because the current collection breaks the C3 symmetry. To observe the C3 pattern, the current collection direction should rotate together with the light polarization, either in the parallel configuration (current collection is parallel to light polarization) or perpendicular (current collection is perpendicular to light polarization) configuration, similar to the standard SHG measurement. It is not clear if this was the case.
5. The authors need to provide evidence (e.g., SHG, Raman) to support the alignment of the TMD and magnetic layer in the system.

Minor concerns:

1. Could the authors explain why the open circuit voltage for different powers in Fig. 2c remains almost the same? What does this indicate?
2. I noticed an unfortunate bubble right in the middle of the channel (Fig. 2a). How can the authors be sure that the observed polarization-dependent current in Fig. 2b and Fig. 2f is not an artifact caused by the bubble?
3. Some sentences are very hard to follow (e.g., lines 134-136). The authors should improve their overall presentation.
4. The authors mention a WSe2 sample in Fig. 1e without much description. What is the purpose of involving WSe2?

Reviewer #4

(Remarks to the Author)

The manuscript "Nonlinear Photovoltaic Effects in Monolayer Semiconductor and Layered Magnetic Material Hetero-Interface with P- and T-Symmetry Broken System" by Shuichi Asada and colleagues investigated the photocurrent generated in a MoS2/CrPS4 heterostructure system with different magnetic states of CrPS4 and, consequently, different photocurrents. Studying bulk photovoltaics in magnetic systems is certainly interesting and timely. Previously, much of the attention was confined to bulk photovoltaics in non-magnetic, non-centrosymmetric systems where the injection current can only be generated by circular polarized light and the shift current can only be generated by linear polarized light. By breaking time-reversal symmetry, the injection current can be generated by both linear and circular light through different mechanisms, and the shift current can also be generated by both linear and circular light through different mechanisms. This is an interesting area to explore both theoretically and experimentally. In the paper, the authors claimed the observation of magnetic injection due to linear polarized light, which is the main novelty of this work. However, the current evidence and presentation do not clearly support this claim.

Major concerns:

1. As mentioned above, since the system simultaneously breaks P and T below the Néel temperature, linearly polarized light can generate both shift and magnetic injection currents. In other words, both effects are allowed to exist in the magnetic phase with linear light. It is not clear how the authors rule out the possibility of shift currents for the observed photocurrent. The observation in Fig. 4, showing the change in photocurrent with magnetic field, is not strong evidence of magnetic nature of the photocurrent in my opinion, as the resistance of the sample could change with different magnetic states, potentially inducing changes in photocurrents. Since magnetic injection current is the main novelty of this work, the authors need to provide strong evidence and explanations to substantiate this claim.

2. Related to the above question, one of the important pieces of evidence for magnetic injection should be its close relation with the magnetic structure. To me, the most important data to show the magnetic origin is the comparison between Fig. 3b and Fig. S7b, where the authors claim that when the spin configuration of the topmost layer of CrPS4 becomes opposite, the photocurrent also reverses. My concern is that this comparison is made between two different devices that were only conjectured to have opposite spin configuration without independent evidence. Why can't the authors change the spin configuration with an external B field? Relatedly, can the authors show forward and backward magnetic field scans, and is there a hysteresis? If the relation between the magnetic structure and photocurrent can be established experimentally, can the authors explain better how the top-layer spin of CrPS4 changes the injection current?

3. What is the temperature for the measurement presented in Figure 2? It seems to be in the non-magnetic phase above 40 K. In this non-magnetic regime, is there any difference of the claimed effect from a previous paper (Science 372, 68–72, 2021) which reported the observation of a large shift current in WSe₂ + black phosphorus, where an in-plane polarization is created at the interface?

4. The description and analysis of the polarization dependence in Fig. 2, starting in line 160, are confusing. Particularly, how was Fig. 2f measured? Did the authors fix the current collection direction or not? If the current is only collected along a fixed direction while changing the polarization, then even for a system with out-of-plane C₃ symmetry, the polarization dependence should be two-fold instead of three-fold because the current collection breaks the C₃ symmetry. To observe the C₃ pattern, the current collection direction should rotate together with the light polarization, either in the parallel configuration (current collection is parallel to light polarization) or perpendicular (current collection is perpendicular to light polarization) configuration, similar to the standard SHG measurement. It is not clear if this was the case.

5. The authors need to provide evidence (e.g., SHG, Raman) to support the alignment of the TMD and magnetic layer in the system.

Minor concerns:

1. Could the authors explain why the open circuit voltage for different powers in Fig. 2c remains almost the same? What does this indicate?

2. I noticed an unfortunate bubble right in the middle of the channel (Fig. 2a). How can the authors be sure that the observed polarization-dependent current in Fig. 2b and Fig. 2f is not an artifact caused by the bubble?

3. Some sentences are very hard to follow (e.g., lines 134-136). The authors should improve their overall presentation.

4. The authors mention a WSe₂ sample in Fig. 1e without much description. What is the purpose of involving WSe₂?

Version 1:

Reviewer comments:

Reviewer #1

(Remarks to the Author)

In general I am satisfied with the authors's changes, and I believe the scientific content of the work is clearly suitable for Nat. Comm. As a comment on nomenclature, especially regarding the response to the other referees: the injection current is one type of ballistic current. In theory, ballistic current means it comes from diagonal contributions to the density matrix (more physically from the difference of intraband velocities between electrons and holes) while shift current comes from off-diagonal contributions (from interband velocity matrix elements). Even with time-reversal symmetry T, there can be ballistic currents with linear polarization, but they must come from scattering (i.e. phonons, e-e interactions). If those are neglected, then the only ballistic current is the injection CPGE. It is in this sense that after breaking T a ballistic injection current occurs with linear polarization.

Note: The abstract still contains the wording "parallel to P-symmetry broken axis" which is meaningless. Can the authors explain what this means? Do they mean parallel to the mirror plane?

Reviewer #2

(Remarks to the Author)

Authors significantly improved their manuscript with detailed explanation of their work. I would recommend this version of the manuscript to be published in Nat. Commun..

Reviewer #3

(Remarks to the Author)

After reviewing the authors' responses, it is evident that they have made some improvements to the manuscript. However, I'm not fully convinced regarding the physical mechanisms presented, and I hope the authors can think more about the below questions and provide more convincing answer or make some adjustments in the paper.

1. The physics and technology behind the first two figures are essentially identical to the previous WSe₂/BP work. The authors should emphasize the unique contributions of their paper rather than dedicating half of the contents to what has already been shown in previous works.
2. As the authors are discussing all the phases in the system (FM, cAFM, AFM, non-magnetic), they should either clearly explain the mechanism behind each state or just reduce the scope of the topic. In system with inversion and time-reversal breaking symmetry, linear shift current, circular injection current, linear injection current and circular shift current all co-exist. The author needs to explicitly define and prove what are the mechanisms for a given temperature and magnetic field/magnetic state for the system. Also, when two spin states are time-reversal partners, the photocurrents tied to those states should also reverse. Several questions regarding on the topic:
 - 2.1 For Fig. S14, what are the magnetic phases when the field is sweeping up and down to around 3T? Why there is a hysteretic opening?
 - 2.2 For Fig. 4c, the two c-AFM states are time-reversal partners. Why are the photocurrent there identical?
 - 2.3 if the author could not determine the AFM domain, can they make a MoS₂-CrPS₄-MoS₂ sandwich structure and show top/bottom MoS₂ have same or different behavior?
3. For the newly added paragraph (line 166-179), I don't understand the logic of "however" and "thus" at line 168 and 171. Are the authors implying the system should have C3 in line 168? And how do they conclude that the current is shift current by comparing them?
4. There are still several grammar and spelling errors

Reviewer #4

(Remarks to the Author)

Version 2:

Reviewer comments:

Reviewer #3

(Remarks to the Author)

I am satisfied with the reply and revisions and support its publication in Nature Communications.

Reviewer #4

(Remarks to the Author)

Manuscript code: NCOMMS-24-26315-T

Title: Nonlinear photovoltaic effects in monolayer semiconductor and layered magnetic material hetero-interface with P - and T - symmetry broken system

Figure updates:

1. We have defined the (x, y, z) axes for the crystal structure and added it to Fig. 1a.
2. The y -axes of Fig. 4a and b were changed to show the both positive and negative values.
3. We also added additional experiment results of I - V curves of monolayer MoS₂ in Fig. S2.
4. We have conducted additional photocurrent measurements in bulk CrPS₄ device with different electrode materials, and added as the new data in Fig. S3.
5. The polar plot of polarization-dependent spontaneous photocurrent with another MoS₂/CrPS₄ heterostructure device has been added in Fig. S6.
6. We have added the new photocurrent measurement data of MoS₂/CrPS₄ heterostructure device with the circular polarized light in Fig. S7.
7. The temperature dependence of fill-factor from the I - V curves in the MoS₂/CrPS₄ heterostructure device was shown in Fig. S8.
8. We have also conducted the low-temperature photocurrent measurement in bulk CrPS₄ device, and the results for I - V curves and the spontaneous photocurrent comparing with those in MoS₂/CrPS₄ heterostructure device were shown in Fig.S13.
9. The spontaneous photocurrent hysteresis of MoS₂/CrPS₄ heterostructure with sweeping the external magnetic field was shown in Fig. S14.
10. The R and V_{OC} of MoS₂/CrPS₄ heterostructure device as a function of external magnetic field at 35 K were calculated from the I - V data as shown in Fig. 4b, and these calculated results were shown in Fig. S15.
11. The polar plot of polarized Raman scattering intensity as a function of angle of linearly polarized light have been added in Fig. S16.
12. The SHG measurement result and optical image of MoS₂ monolayer, CrPS₄ bulk and MoS₂/CrPS₄ heterostructure device have been added in Fig. S17.

Summary of changes:

13. We have added the detail explanation of shift and injection current, their responses to linear and circular polarization, and their magnetic odd properties with respect to T -symmetry in the introduction part of manuscript.
14. We have added the references according to Reviewer's suggestions.
15. We have defined new (x, y, z) axes for crystal structure to explain the direction of measurement more clearly.
16. We have added the discussion about the competition of shift current and ballistic current according to the Reviewer's comments.

17. The origin of spontaneous photovoltaic effect was discussed and compared with the case of a device only with monolayer MoS₂ or bulk CrPS₄.
18. The stacking angle θ was defined from the results of polarized Raman scattering and SHG measurement.

Point by point response to reviewers:

We are thankful for your helpful comments and elaborated suggestions. We have considered the Reviewers' comments carefully to further improve our manuscript for the publication in *Nature Communications*.

The revisions have been made to the manuscript following the Reviewers' comments. We hope that we have now produced a better account for our work. We would like to resubmit our revised manuscript to *Nature Communications* for publication. We studied the Reviewers' comments carefully and the revisions made are described as follows. We introduced additional data following the advices received from the Reviewers.

Reply to the comments of Reviewer #1

In this work, the authors report the experimental demonstration of a magnetically switchable linear photogalvanic effect (LPGE) in an interface material. The authors have performed all the standard checks to demonstrate that this is a PGE (linears on power, polarization dependence, spot position dependence to eliminate contact effects), as well as clear and sharp dependence on the magnetization state. There has been great interest in the community to show the existence of this effect, and in my opinion this proof alone would be merit for publication.

However, I must also emphasize that the presentation, organization and motivation of the manuscript could use a very significant improvement. Below the authors can find a list of recommendations to ensure the paper can be accepted for publication and has broader impact in the community.

We would like to thank the Reviewer for careful reading and summary of our manuscript appropriately. We considered the Reviewers' helpful suggestions and improved the manuscript. Our answers and revisions are described below.

1. The authors need to explain the difference between shift and injection currents more clearly in the introduction. The key difference is that in non-magnetic systems, shift current gives linear PGE, while injection gives only circular PGE. In magnetic systems injection provides an extra LPGE, which is what the authors are observing here. In addition the shift current might change too after the magnetic transition, but the linear injection current is odd under magnetization change, as the authors also observe.

Thank you for your helpful comments. According to the Reviewer's advices, we have expanded the explanation of the difference between shift current and injection current in the introduction part of the main-text. The particular attention was paid to non-magnetic systems, and a clear distinction was made between the generation of linear PGE due to shift currents and the generation of circular PGE due to injection currents. Moreover, the descriptions about the injection current producing additional LPGE in the magnetic systems were added. We have also described the possible changes in the shift current after the magnetic transition and the odd nature of the linear injection

current under the magnetization change, and clarified the relation with the results observed in this study.

2. The authors should compare their results with Song, *Sci. Adv.* **7, eabg8094 (2021), where switchable photocurrents were reported in CrI₃. What is the difference between these works, and how does this work represent an improvement? The authors only provide two references on magnetic injection current which do not appear in the reference list (16-17). Presumably these are (18-19) which are repeated twice. I think it would be fair to cite the original works on shift current like Belinicher *Sov. Phys. Usp* **23**, 199 (1980), Sipe *PRB* **61**, 5337 (2000), as well as the recent ones on magnetic photocurrents Fei *PRB* **102**, 035440 (2020), Watanabe *PRX* **11**, 011001 (2021).**

Thank you for the advices regarding the citations. The citations have been revised according to the Reviewer's advice. Moreover, we have added the appropriate references and cited the references as the Reviewer suggested with a relation to the shift current and magnetic photocurrent.

The Reviewer also requests to compare the result [T. Song *et al.*, *Sci. Adv.* **7**, eabg8094 (2021)] in an antiferromagnetic (AFM) material of CrI₃, where each layer shows ferromagnetic ordered in the out-of-plane direction, similar to that in CrPS₄. However, the results show the induced current in the *out-of-plane* direction, which differs from that the in-plane direction in *P*-symmetry. In addition, the paper uses a very thin layer sample and shows that the spontaneous current is inverted depending on the spin direction and that the spontaneous current is very small in the antiferromagnetic state without an external magnetic field. This might be due to the inversion of the current direction depending on the spin direction (derived from the fact that it is an odd function), as indicated by Y. Zhang *et al.*, *Nat. Commun.* **10**, 3783 (2019), which cancels the inverted magnetic photocurrent in the AFM state.

In our study, however, the magnetic photocurrent generation was observed even in the AFM state. This causes due to artificial *P*- and *T*-symmetry breaking at the hetero-interface, which is unique and significant different from the previous studies.

3. The authors need to clarify their discussion about symmetry by using standard names for symmetry operations, and by using only one name for each symmetry. There is no such thing as an “axis of inversion symmetry”. The inversion operation $(x,y,z) \rightarrow -(x,y,z)$ is done with respect to a point, called inversion center. When only one coordinate is inverted $(x,y,z) \rightarrow (-x, y,z)$ this is called a mirror plane. The authors later refer to the aligned vertical mirror planes of the heterostructure, which is the correct name. Inversion symmetry is always broken at every interface, but this does not necessarily generate a polar axis, as the authors seem to imply. To clarify the discussion, the authors should draw standard coordinate axis, define the symmetry operations with respect to that axis, and explain the symmetry constraints on the photocurrent. The sentences “parallel to the spatial inversion symmetry broken axis at the interface”, “uni-axis of inversion”, “inversion... is maintained only in the uniaxis” should be removed.

Thank you for your helpful comments. According to the Reviewer's suggestion, we have revised the description of symmetry throughout the paper. We updated Fig. 1 to define (x, y, z) axes to make the discussion of symmetry operations more clearly. Based on this newly-defined coordination, each symmetry operations (inversion center, mirror plane) were defined in a rigorous manner. We have added detailed explanations of the polarity induced by these symmetry operations using the coordinate axes. Moreover, the description of spatial inversion symmetry breaking has been re-expressed precisely using the defined coordinate axes. The inappropriate expressions such as “inversion symmetry axis” have been removed, and the changes in symmetry at the interface have been described.

4. The authors say they show dI/dV "along the expected polarization direction", what polarization is this? Can the authors please use coordinate axes to describe this? Is there any proof that there is charge polarization in the plane, and not out of plane or anywhere in between?

Thank you for your helpful comments. The predicted direction of polarization corresponds to the direction along the x -axis based on newly-defined coordinate axes, as indicated by the red arrow in Fig. 1a. The polarization is induced due to the hetero-interface of materials with different rotational symmetries around the z -axis, which has been discussed in the previous paper in the similar system of WSe_2/BP [T. Akamatsu, *et al.*, *Science* **372**, 68 (2021)]. For out-of-plane, the charge polarization may exist between MoS_2 and $CrPS_4$ due to their distinctly different structures. However, in this study, since the electrodes are attached to the monolayer MoS_2 surface in the in-plane direction, we believe that the effect of electron transfer due to charge polarization in the out-of-plane direction is negligible.

5. Other use of language: Threefold-symmetry MoS_2 -> Threefold symmetric MoS_2 . "Time reversal symmetry broken system" -> System with broken time-reversal symmetry. Avoid repeating parenthesis in inversion (P -) and time reversal (T -) throughout the text. Avoid use of parenthesis as in "The increase (decrease) in photocurrent is caused by a decrease in the negative (positive) current", which make the text harder to read.

According to the Reviewer suggestions, we have revised the use of language and parenthesis. To make the text more readable for the readers, we have updated the manuscript to remove unnecessary notations and repetitions.

6. The manuscript contains unnecessary repetition, for example in the description of what the shift current is which appears several times in the text.

According to Reviewer's helpful advice, we have removed unnecessary explanations and modified the main-text to make it clear.

7. In my opinion, the authors use the word "topological" without need. It plays no role in the explanation and might as well be removed.

We described the shift current as a “topological” current according to the convention,

but we do not discuss about the topological origin in this study. Thus, we have removed this unnecessary word from the manuscript and updated the text to be more appropriate for the discussion.

8. The authors appear to use normal incident light but this is not stated anywhere in the text.

Thank you for your helpful comment. We have added more detailed information in the manuscript about the light conditions during the measurements.

Reply to the comments of Reviewer #2

Photovoltaic effects in P - and T -symmetry broken systems were reported by Asada et al., which is intriguing and fundamentally promising in the field. At a specific temperature, the combined effect of shift and injected current is correlated with the observed PV effect. It thus demonstrates the need for the P - and T -broken system. However, there are a few important aspects that need to be addressed before publication.

We would like to appreciate the Reviewer's careful reading of our manuscript. We have discussed the origin of the observed spontaneous photocurrents in more detail based on the Reviewer advice, and have conducted additional many experiments to determine that they come from the shift current and the magnetic photocurrents. We believe that our new experiments and discussions have satisfactorily resolved the important aspects necessary for publication.

1. The authors solely focused on the Shift current response when examining the overall photovoltaic effect. The bulk photovoltaic effect is a result of the combined response of ballistic and shift current. It's unclear why the Authors overlooked the ballistic contribution.

As the Reviewer pointed out, the bulk photovoltaic effect would include a contribution from the ballistic currents. In particular, the effect of ballistic currents is predicted to be larger in the low-dimensional materials [Z. Dai and A. M. Rappe, *Phys. Rev. B* **104**, 235203 (2021)], and the contribution of ballistic currents will not be completely zero. We have performed additional experiments to discuss this important issue. From the additional experimental results, we clearly show that the shift current is considerably larger than the ballistic current in the MoS₂/CrPS₄ heterostructure used in this study. The detailed experiments and results will be presented in response to the questions as replies to the next comment.

2. The light polarisation angle-dependent ISC does not verify materials exhibit shift current response. It is the characteristic of the bulk PV phenomenon. It is also applicable to the non-linear current versus intensity characteristics. It could be due to competition between the Ballistic and Shift current response. The authors can determine the relaxation time of the carriers in order to determine the contribution that solely originates from the Shift current response. The band alignment in the P and T -symmetry broken system can also shed light on the mechanism.

Thank you for your important comment. The Reviewer's comments are exactly right: the light polarization angle-dependent ISC and the non-linear current versus intensity characteristics are properties of the bulk PV phenomenon, and these do not rule out a ballistic current response. As described above, the relaxation time of carriers are one of the information to distinguish the ballistic and shift current, however, it difficult to measure in our experimental setup. Therefore, we performed the PV measurements using circularly polarized light (CPL) in addition to linearly polarized light (LPL) to directly estimate the magnitudes of the shift and ballistic currents. In the previous study

that mainly discussed the ballistic currents, it was shown that the spontaneous photocurrent by the LPL shows the competition between the ballistic and shift current, while the purely ballistic current is only induced by the CPL [A. M. Burger *et al.*, *Sci. Adv.* **5**, eaau5588 (2019), V. I. Belinicher *et al.*, *Zh. Eksp. Teor. Fiz.* **83**, 649–661 (1982)]. The disappearance of the shift current excited by CPL is also consistent with the theoretical understanding [H. Wang, X. Qian, *npj Comput Mater* **6**, 199 (2020)].

We have conducted the additional experiments in MoS₂/CrPS₄ heterostructure both by LPL and CPL conditions. Figure R1 shows the *I-V* curves under the same laser intensity condition by LPL (red line) and CPL (blue line). The first interesting point is that the spontaneous photocurrents in the reversed signs were observed in the LPL (+ 4 pA) and CPL cases (-185 pA), which are consistent with the change on the sign of ballistic and shift currents that has been reported in the previous study [A. M. Burger *et al.*, *Sci. Adv.* **5**, eaau5588 (2019)]. Moreover, the spontaneous photocurrent by LPL condition is much larger than that by CPL condition, which strongly suggests that the contribution of ballistic current is negligibly small in comparison with the shift current in the MoS₂/CrPS₄ heterostructure used in this study.

Fig. R1 *I-V* curves of MoS₂/CrPS₄ heterostructure device with LPL (red line) and CPL (blue line).

3. Materials show the above bandgap PV effect in accordance with the bulk PV phenomenon, which may allow them to surpass the efficiency S-Q limit. The observed V_{OC} in the system is nowhere close to the above bandgap voltage. Please comment on this. From here, one might think that this could simply be a contribution to the interface. How do authors eliminate this contribution?

Thank you for your helpful comments. As the Reviewer pointed out, the open-circuit voltage is indeed not limited by the band gap of material, which may allow to surpass the efficiency of S-Q limit in the bulk PV phenomena. However, the bulk PV phenomena due to shift current do not necessarily indicate an open circuit voltage

higher than the band gap of the material [S. Aftab et al., *Laser Photonics Rev.* **17**, 2200429 (2023)]. We think that the series resistance and spontaneous current are not large enough to provide a large open-circuit voltage in the MoS₂/CrPS₄ heterostructure.

The spontaneous photocurrent of the MoS₂/CrPS₄ heterostructure as shown in Fig. R2 (red curve) suggests not from the simple interface such as monolayer MoS₂. We also conducted the *I-V* measurement of monolayer MoS₂ for comparison. The spontaneous photocurrent of monolayer MoS₂ in Fig. R2 (blue curve) could not be observed, which is well consistent with the previous result [Y. Dong et al., *Nat. Nanotech.* **18**, 36-41 (2023)].

Fig. R2 *I-V* curves of MoS₂/CrPS₄ heterostructure and monolayer MoS₂ device by LPL with the power of 600 μW. Inset shows the optical image of monolayer MoS₂ device.

4. Increasing current response below 40 K could also be a consequence of decreasing resistance in the materials. From Fig. 3a, it is also clear that decreasing the temperature follows the decreasing ISC trend by compensating for increasing V_{OC}. What about the trend in Fill Factor? What is the trend of R and V_{OC} response in various magnetic fields?

Thank you for your helpful comment. We think that the current response below 40 K does not come from a decrease in material resistance, because it would not explain the constant spontaneous photocurrent above 40 K, even though the change of resistance as a function of temperature in Fig. 3b. Figure R3 shows the temperature dependence of fill-factor calculated from the *I-V* curves. The fill-factors show the constant values of about 0.25 independent on the temperature.

Fig. R3 Temperature dependence of fill-factor from the I - V curves in the MoS₂/CrPS₄ heterostructure device. The dotted line is a guide for eyes.

The responses of R and V_{oc} at 35 K are shown in Fig. R4 from the magnetic I - V measurements in Fig. 4 of main text, where the R and V_{oc} are calculated using linear fitting procedures. Fig. R4a shows that R does not change significantly with external magnetic field, while V_{oc} increases due to the effect of change in the spontaneous current. These results indicate that the transition of the magnetic state due to the external magnetic field induces a change in the spontaneous current, not in the resistance.

Fig. R4 (a) R and (b) V_{oc} as a function of external magnetic field.

5. It is crucial to provide the ferroelectric response at the interface for different temperatures.

Thank you for your helpful comment. The crystal structure and broken P -symmetry at the hetero-interface are very important for the ferroelectric response in this

heterostructure device, however the ferroelectric response cannot be directly observed in the similar heterointerface such as WSe₂/BP [T. Akamatsu *et al.*, *Science* **372**, 6537 (2021)] and so on, due to the experimental difficulties. The *out-of-plane* ferroelectric response has been indicated by the spatially resolved measurement of *I-V* response by conductive-AFM [M. Lv, *et al.*, *Adv. Mater* **34**, 2203990 (2022), and W. Wu, *et al.*, *Phys. Rev. Lett.* **104**, 217601 (2010)]. However, it is difficult to apply this technique to the *in-plane* ferroelectric response in the hetero-interface devices based on in-plane 2D material system, because it is not practical to have the needle control and touch to the sides of the atomic layer. The experimental fact that the spontaneous photocurrent at 10 K differs between the MoS₂/CrPS₄ hetero device and CrPS₄ bulk device might be one of the proofs of ferroelectric response in the hetero-interface at low-temperature as shown in Fig. S13. Moreover, we have carefully discussed the distortion of the crystal structure from optical measurements, and the reported lattice constant especially below the magnetic transition temperature of 40 K, as shown in Fig. S9-S11. In conclusion, we believe that the crystal structure remains unchanged from room temperature to below the transition temperature, and that the ferroelectric response might also be unchanged.

6. Without MoS₂, CrPS₄ is a *T*-symmetry broken system. What is the behaviour of temperature-dependent injection current in only a *T*-symmetry broken system? Does it show injection current? Does it depend on the circular polarisation angle?

Thank you for your helpful comment. To answer the Reviewer's question, we measured the *I-V* characteristics of CrPS₄ at various temperatures as the additional experiments. Figure R5 shows the *I-V* curves of CrPS₄ bulk excited by RCPL (Right-hand circularly polarized light) and LCPL (Left-hand circularly polarized light) and MoS₂/CrPS₄ heterostructure excited by LPL (Linearly polarized light) at room temperature under a constant light power condition of 0.92 mW. The spontaneous photocurrent is observed only in MoS₂/CrPS₄ heterostructure, which suggests that the injection current introduced by CPL in the nonmagnetic state of CrPS₄ is negligibly small. In the CPL measurements, the laser light was incident normal to the device. The circular polarization angle dependence was not measured, because CrPS₄ does not show injection current. Moreover, the resistance of CrPS₄ bulk is much larger than that in MoS₂/CrPS₄ heterostructure from the gradient of *I-V* curves.

We also prepared bulk CrPS₄ devices made with different electrode materials and performed *I-V* measurements at room temperature. However, as shown in the inset of Fig. R5, the spontaneous photocurrent is not observed in bulk CrPS₄ devices with different contact metals (Au/Cr, and Au/Bi). These results of only bulk CrPS₄ and monolayer MoS₂, which are component materials of heterostructure do not show the spontaneous photocurrent as shown in Fig. R2 and R5, however those of heterointerface of MoS₂/CrPS₄ clearly show the spontaneous photocurrent. These results clearly suggest that the artificial *P*-symmetry breaking at the hetero-interface with MoS₂ and CrPS₄ induces the spontaneous photocurrent, which are consistent with the previously reported results of WSe₂/BP heterointerface [T. Akamatsu, *et al.*, *Science* **372**, 68 (2021)].

Fig. R5 I - V curves of bulk CrPS₄ under CPL and MoS₂/CrPS₄ heterostructure under LPL excitation conditions at room temperature. The inset shows I - V curves of MoS₂/CrPS₄ hetero and bulk CrPS₄ device with different electrode materials.

We further conducted the I - V measurements of bulk CrPS₄ excited by LPL in a T -symmetry-breaking system at 10 (below T_N) and 300 K (above T_N). Figure R6 shows that no spontaneous photocurrents are observed in the bulk CrPS₄ even in the antiferromagnetic state below T_N , where only the T -symmetry breaking occurs in the bulk CrPS₄. It is theoretically understood that both shift current and magnetic injection current require P -symmetry breaking [H. Wang and X. Qian, *npj Comput. Mater.* **6**, 199 (2020)], suggesting that artificial P -symmetry breaking at the heterointerface with monolayer MoS₂ in MoS₂/CrPS₄ heterostructure plays a very important role in both shift current and linear injection current below the magnetic phase transition temperature of CrPS₄.

Fig. R6 (a) I - V curves of bulk CrPS₄ and MoS₂/CrPS₄ heterostructure device at 10 K (below T_N) and 300 K (above T_N), excited by LPL. (b) Temperature dependence of spontaneous photocurrent of MoS₂/CrPS₄ heterostructure and bulk CrPS₄ device.

7. The magnitude of the current response in the inset of Fig. 3b (upper panel) is quite high. It is not clear how the experiment is performed. Is it measured with bias? It is mentioned that it is measured in different devices. How can similar device architecture display significantly different current values with opposite trends? Authors should address this behaviour.

Thank you for your helpful comment. The both results in Fig. 3b and its inset show the spontaneous currents derived from I - V measurements without the bias-voltage, and the similar light intensity of 0.60 and 0.68 mW. The difference of spontaneous photocurrent itself might come from the device characteristics such as the channel length, contact resistance between the channel material and electrode, and so on according to the previous report [M. Nakamura et al., *Appl. Phys. Lett.* **113**, 232901 (2018)]. We think that the significantly different current values with opposite trends are not related the value of spontaneous current.

Minor comments:

1. What is the intensity of the light used in PV experiments?

Thank you for your comment. The illuminated laser powers in Fig. 3 and its inset are the almost similar condition of about 0.60, and 0.68 mW, respectively. In the magnetic measurement as shown in Fig. 4, the light intensity was about 1.0 mW. Each light intensity was added to the manuscript.

2. What would be expected PV outcomes when the magnetic field is parallel to the polarization direction? Do the authors expect any change in the overall response?

As pointed out by the Reviewer, the spontaneous photocurrent under the parallel magnetic field to the polarization direction would be important issues. The observed magnetic photocurrent is caused by both P - and T -symmetry breaking; the P -symmetry

is broken at the heterostructure interface, independent of the spin state, so we would expect the magnetic photocurrent to occur even when the magnetic field is parallel to the polarization direction. However, these might be included in the future studies.

3. Experimental data in the Fig. 4a and b are not smooth. Also, the scale in the y-axis is plotted only in the positive value. It is important to show full-scale reading.

Thank you for your comment. The I - V measurements with an applied high magnetic field using a superconducting magnet in Fig. 4a has been conducted by the different measurement system from those in Fig. 3. In the I - V measurement system under magnetic-field, the several factors arising from the vibration, less-electrical shielding cause the additional noise for the data, even though we can see the significant changes of I - V curves as a function of magnetic field. According to the Reviewer advices, the data in different scale are shown in Fig. R7.

Fig. R7 I - V (a) and (b) characteristics of MoS₂/CrPS₄ heterostructure device with external out of plane magnetic field from -7 to 7 T at 10 and 35 K, respectively.

4. Page-5, “The I - V characteristics under dark conditions exhibit simple linear behavior...to the usual photoconductive effect”. How I - V response in the dark can be related to the photoconductive effect?

Thank you for the comment. As pointed out by the Reviewer, there is no direct relationship between I - V response under dark conditions and the photoconductive effect. Here, we simply explain the fact that the current with bias voltage under dark conditions was not zero. To avoid the misunderstanding, we changed the description in the revised manuscript.

5. V_{OC} obtained in the magnetic PV measurements is nearly one order of magnitude higher than the normal I - V measurements presented in Fig. 3. Why are the measurements inconsistent?

Thank you for your helpful comment. We have added the detailed measurement conditions of magnetic PV measurements in the manuscript. As described above, the value of V_{OC} depends on the characteristics of devices such as channel length, contact

resistance, and so on, as similar to that of spontaneous photocurrent (I_{sc}). As further steps in the future, we will optimize the PV parameters of V_{oc} , and I_{sc} by reduction of the variations of these values among the device and device.

Reply to the comments of Reviewer #3

The manuscript "Nonlinear Photovoltaic Effects in Monolayer Semiconductor and Layered Magnetic Material Hetero-Interface with *P*- and *T*-Symmetry Broken System" by Shuichi Asada and colleagues investigated the photocurrent generated in a MoS₂/CrPS₄ heterostructure system with different magnetic states of CrPS₄. They use temperature and magnetic field as tuning knobs to access different magnetic states of CrPS₄ and, consequently, different photocurrents. Studying bulk photovoltaics in magnetic systems is certainly interesting and timely. Previously, much of the attention was confined to bulk photovoltaics in non-magnetic, non-centrosymmetric systems where the injection current can only be generated by circular polarized light and the shift current can only be generated by linear polarized light. By breaking time-reversal symmetry, the injection current can be generated by both linear and circular light through different mechanisms, and the shift current can also be generated by both linear and circular light through different mechanisms. This is an interesting area to explore both theoretically and experimentally. In the paper, the authors claimed the observation of magnetic injection due to linear polarized light, which is the main novelty of this work. However, the current evidence and presentation do not clearly support this claim.

Thank you for your important and helpful comments. Following the Reviewer's advices, an additional experiments and a more in-depth discussion has provided in the revised manuscript.

Major concerns:

1. As mentioned above, since the system simultaneously breaks *P* and *T* below the Néel temperature, linearly polarized light can generate both shift and magnetic injection currents. In other words, both effects are allowed to exist in the magnetic phase with linear light. It is not clear how the authors rule out the possibility of shift currents for the observed photocurrent. The observation in Fig. 4, showing the change in photocurrent with magnetic field, is not strong evidence of magnetic nature of the photocurrent in my opinion, as the resistance of the sample could change with different magnetic states, potentially inducing changes in photocurrents. Since magnetic injection current is the main novelty of this work, the authors need to provide strong evidence and explanations to substantiate this claim.

Thank you for your helpful comments. It is known that the shift current does not show significant changes due to the resistance change of the materials [M. Nakamura *et al.*, *Appl. Phys. Lett.* **113**, 232901 (2018)]. We clearly showed the constant spontaneous photocurrents independent even in the large change of resistance above 40 K in Fig. 3a. Moreover, the results of *I-V* measurements also show that the resistance of MoS₂/CrPS₄ heterodevice does not change significantly with varying the magnetic fields, as shown in Fig. S8. Therefore, the magnetic field dependence of the photocurrent shown in Fig. 4 provides strong experimental evidence for the magnetic injection current.

2. Related to the above question, one of the important pieces of evidence for magnetic injection should be its close relation with the magnetic structure. To me, the most important data to show the magnetic origin is the comparison between Fig. 3b and Fig. S7b, where the authors claim that when the spin configuration of the topmost layer of CrPS₄ becomes opposite, the photocurrent also reverses. My concern is that this comparison is made between two different devices that were only conjectured to have opposite spin configuration without independent evidence. Why can't the authors change the spin configuration with an external B field? Relatedly, can the authors show forward and backward magnetic field scans, and is there a hysteresis? If the relation between the magnetic structure and photocurrent can be established experimentally, can the authors explain better how the top-layer spin of CrPS₄ changes the injection current?

As the Reviewer pointed out, the detailed spin states of CrPS₄ including the spin-up (-down) in the top layer cannot be directly observed in this study. The spin-state of CrPS₄ changes from AFM to CAFM (FM) by applying an external magnetic-field, rather than an inverted AFM state with top layer spin-flipping. Thus, the AFM state with opposite spin configurations were selected using the different device.

We also measured hysteresis loop for the spontaneous photocurrent. Figure R8 clearly shows the hysteresis of spontaneous photocurrent under forward and backward magnetic field scan, which also strongly supports the magnetic injection current depending on the magnetic spin-configuration of CrPS₄ in MoS₂/CrPS₄ heterostructure device.

Fig. R8 Spontaneous photocurrent hysteresis of MoS₂/CrPS₄ heterostructure with sweeping the external magnetic field from 0 to 7 T (red circles), and 7 to 0 T (blue triangles), respectively.

3. What is the temperature for the measurement presented in Figure 2? It seems to be in the non-magnetic phase above 40 K. In this non-magnetic regime, is there any difference of the claimed effect from a previous paper (Science 372, 68-72,

2021) which reported the observation of a large shift current in WSe₂ + black phosphorus, where an in-plane polarization is created at the interface?

Thank you for your helpful comment. The measurement condition in Fig. 2 is in room temperature (300 K), which corresponds to non-magnetic phase. Also, the results of shift current due to the induced in-plane polarization in the MoS₂/CrPS₄ interface of nonmagnetic region is similar to the previous paper presented by the Reviewer.

4. The description and analysis of the polarization dependence in Fig. 2, starting in line 160, are confusing. Particularly, how was Fig. 2f measured? Did the authors fix the current collection direction or not? If the current is only collected along a fixed direction while changing the polarization, then even for a system with out-of-plane C₃ symmetry, the polarization dependence should be two-fold instead of three-fold because the current collection breaks the C₃ symmetry. To observe the C₃ pattern, the current collection direction should rotate together with the light polarization, either in the parallel configuration (current collection is parallel to light polarization) or perpendicular (current collection is perpendicular to light polarization) configuration, similar to the standard SHG measurement. It is not clear if this was the case.

Thank you for your helpful comments. In the measurement in Fig. 2f, the current collection direction was fixed and only the light polarization direction was rotated. This relationship between C₃ symmetry and polarized spontaneous photocurrent has been already discussed in the previous paper [T. Akamatsu *et al.*, *Science* **372**, 6537 (2021)]. As pointed by the Reviewer, the system with C₃ symmetry is expected to show two-fold pattern theoretically [V. M. Fridkin, *Gordon and Breach Science Publishers* (1992)]. According to the theoretical prediction, the system with C₃ symmetry exhibits the positive and negative spontaneous photocurrent depending on the direction of polarization. However, the experimentally observed spontaneous photocurrent always show the positive value in the previous paper [T. Akamatsu *et al.*, *Science* **372**, 6537 (2021)] and also in this study, which has not been fully understood theoretically at the present stage. We have updated the manuscript appropriately so that these relationships can be more clearly understood about this point.

5. The authors need to provide evidence (e.g., SHG, Raman) to support the alignment of the TMD and magnetic layer in the system.

Thank you for your helpful suggestion. According to the Reviewer advices, we have conducted additional experiments of SHG and polarized Raman scattering to provide the alignment of TMD and magnetic material in addition to information on the edge angles in the optical image. Figure R9a shows polarized Raman scattering spectrum of MoS₂/CrPS₄ heterostructure with changing the linearly polarized excitation light. Moreover, the optical image of measured heterostructure with corresponding relative small angle of 4° between MoS₂ and CrPS₄ is shown in the inset.

The several Raman peaks of MoS₂ and CrPS₄ are clearly observed in Fig. R9a. The polar plot of Raman peak at 306 cm⁻¹ of CrPS₄ shows a clear two-fold symmetry with an axis at angle of 15 ± 5°. Moreover, the peak at 406 cm⁻¹, the overlap of the A_{1g} peak

of MoS₂ and the A-type peak of CrPS₄ also shows slightly distorted ellipse with an axis at angle of $15 \pm 5^\circ$. Both angles as indicated by the dotted line in Fig. R9 are in good agreement with the mirror plane estimated from the optical image [S. Kim *et al.*, *J. Phys. Chem. C* **125**, 4 (2021), and H. Kim *et al.*, *J. Raman Spectrosc.* **51**, 5 (2020)].

Fig. R9 (a) Raman scattering spectrum of MoS₂/CrPS₄ heterostructure. Inset shows the optical image of the measured heterostructure. The relative crystal angles are indicated in the optical image. (b) Polar plot of polarized Raman scattering intensity as a function of angle of linearly polarized light: CrPS₄ A-type peak at 306 cm⁻¹ (red), mixed peak of MoS₂ A_{1g} and CrPS₄ A-type peak at 406 cm⁻¹ (blue).

Figure R10a-c shows the optical images of monolayer MoS₂, bulk CrPS₄ and monolayer MoS₂/CrPS₄ heterostructures, respectively. Each inset shows the schematic of crystal structure, and characteristics crystal axes, as shown in dotted lines. Figure R10d-f show the polar plots of SHG intensity of MoS₂, CrPS₄ and MoS₂/CrPS₄ heterostructures, respectively. The excitation and detection light were linearly polarized, and the SHG signals were detected with a co-linearly polarized configuration (XX) for MoS₂ and MoS₂/CrPS₄ and co-linearly (XX) and cross-linearly polarized configuration (XY) for bulk CrPS₄.

The armchair direction of monolayer MoS₂ inferred from optical image in Fig. R10a well corresponds to the peak of polar pattern of SHG signals in Fig. R10d, where the SHG signal is enhanced in the armchair direction [L. Mennel *et al.*, *APL Photonics* **4**, 3(2019)]. In bulk CrPS₄ as shown in Fig. R10b and e, the longitudinal axis of SHG polar pattern in (XY) configuration is well matched to the <010> crystal direction, which is well consistent with the previous report [D. Hou *et al.*, *Adv. Opt. Mater.* 2400943 (2024)].

In the MoS₂/CrPS₄ heterostructure device as shown in Figs. R10c and f, the polar pattern of SHG signal exhibits six-fold symmetry pattern, which comes from mainly the signals of monolayer MoS₂ in the heterostructure. However, noted that the experimentally observed intensity at two-lobes at 0 and 180 degree show the clearly enhancement in the mirror plane direction of MoS₂/CrPS₄ heterostructure, which strongly suggests that the polarization is induced along this direction [Z. Li *et al.*, *Nat. Commun.* **14**, 5568 (2023)].

These results of optical image, Raman scattering and SHG are consistent with each other, which strongly suggest that the crystal directions of monolayer MoS₂, bulk CrPS₄ and MoS₂/CrPS₄ heterostructure are appropriately evaluated.

Fig. R10 Optical image of (a) MoS₂ monolayer, (b) CrPS₄ bulk, and (c) MoS₂/CrPS₄ heterostructure sample. Each inset shows the schematic of crystal structure, and characteristics crystal axes, as shown in dotted lines. Polar plot of SHG intensity of (d) MoS₂ monolayer, (e) CrPS₄ bulk, and (f) MoS₂/CrPS₄ heterostructure.

Minor concerns:

1. Could the authors explain why the open circuit voltage for different powers in Fig. 2c remains almost the same? What does this indicate?

Thank you for your helpful comment. In Fig. 2c, the reason why the open circuit voltage V_{OC} does not change for different excitation light intensities comes from the compensation of increase of spontaneous photocurrent I_{SC} and decrease of electrical resistance R due to the photoconductive effect, as following the relationship of $V_{OC} = I_{SC} \cdot R$.

2. I noticed an unfortunate bubble right in the middle of the channel (Fig. 2a). How can the authors be sure that the observed polarization-dependent current in Fig. 2b and Fig. 2f is not an artifact caused by the bubble?

Thank you for your helpful comment. The optical image shown in Fig. 2a was taken after all measurements. Figure 11a-c show that the optical image of same sample before device fabrication process, during the photocurrent measurement, and after the measurement, respectively. As clearly seen, the bubble was not present during the measurements. At last, we performed the excitation intensity dependence of spontaneous photocurrent measurements up to much higher intensities conditions (~2 mW) than indicated in Fig. 2c, which might cause some damages. In addition, Figure

S3 shows the results measured on *different* MoS₂/CrPS₄ heterostructure device, which also shows the same results in Fig. 2. The characteristic photocurrent similar to Fig. 2, including nonlinear laser intensity dependence, were also observed in the other devices without bubbles, as shown in Fig. S3. Moreover, as shown in Fig. R12, we observed the similar behaviors of polarization-dependent currents in a different device. Thus, we can conclude that the observed polarization-dependent current in Fig. 2b and 2f is not an artifact caused by the bubble.

Fig. R11 Optical image of the same MoS₂/CrPS₄ heterostructure device as shown in Fig. 2a, (a) before electrode fabrication, (b) during photocurrent measurement, and (c) after measurement.

Fig. R12 (a) Optical image of the different MoS₂/CrPS₄ heterostructure device. (b) Polar plot of polarization-dependent spontaneous photocurrent.

3. Some sentences are very hard to follow (e.g., lines 134-136). The authors should improve their overall presentation.

Thank you for your careful reading of the manuscript. According to the Reviewer's advice, we have re-described and improved the overall presentation of the manuscript.

4. The authors mention a WSe₂ sample in Fig. 1e without much description. What is the purpose of involving WSe₂?

Thank you for your helpful comment. The purpose is to show our experimental results of common behaviors of TMD (MX_2 : $\text{M}=\text{Mo}$, W , $\text{X}=\text{S}$, Se)/ CrPS_4 heterodevice determined by the symmetry. These results clearly show the artificial P -symmetry breaking generates the spontaneous photovoltaic effect both in $\text{WSe}_2/\text{CrPS}_4$ and $\text{MoS}_2/\text{CrPS}_4$ heterodevice.

Base on the helpful comments from the Reviewers, we added a more detailed discussion to our paper and we hope that we have now produced a better account for our work. We would like to express our appreciation to the Reviewers for their suggestions in improving our paper. We will be most grateful if you could offer us a second opportunity to review to *Nature Communications* with the revisions made. Look forward to hearing from you soon.

Yours sincerely,

Kazunari Matsuda

Resubmission to *Nature Communications*

Manuscript code: NCOMMS-24-26315-B

Title: Nonlinear photovoltaic effects in monolayer semiconductor and layered magnetic material hetero-interface with P - and T - symmetry broken system

Figure updates:

1. We have added the summary figure to show the magnetic phase of CrPS₄ as a function of temperature and external magnetic field in Supporting Information.
2. In the caption of Fig. 4(a) and (b), we have added the explanation of plotted current.

Summary of changes:

1. The appropriate correction was made according to the Reviewer's advice in the abstract.
2. We have described the relationship between injection and ballistic current based on the results of I - V measurements with CPL as shown in Fig. S7 at Line 180-184.
3. We have summarized the generation of magnetic injection current and each magnetic phase of CrPS₄ with each temperature and out-of-plane external magnetic field at line 261-262.

Point by point response to reviewers:

We are thankful for your helpful comments and elaborated suggestions. We have considered the Reviewers' comments carefully to further improve our manuscript for the publication in *Nature Communications*.

The revisions have been made to the manuscript following the Reviewers' comments. We hope that we have now produced a better account for our work. We would like to resubmit our revised manuscript to *Nature Communications* for publication. We studied the Reviewers' comments carefully and the revisions made are described as follows. We have added new data of experiments and introduced additional figures following the advices received from the Reviewer.

Reply to the comments of Reviewer #1

In general, I am satisfied with the authors's changes, and I believe the scientific content of the work is clearly suitable for Nat. Comm. As a comment on nomenclature, especially regarding the response to the other referees: the injection current is one type of ballistic current. In theory, ballistic current means it comes from diagonal contributions to the density matrix (more physically from the difference of intraband velocities between electrons and holes) while shift current comes from off-diagonal contributions (from interband velocity matrix elements). Even with time-reversal symmetry T , there can be ballistic currents with linear polarization, but they must come from scattering (i.e. phonons, e-e interactions). If those are neglected, then the only ballistic current is the injection CPGE. It is in this sense that after breaking T a ballistic injection current occurs with linear polarization.

We would like to thank the Reviewer for positive and helpful comments, and are strongly encouraged by these comments. Moreover, we have improved our understanding of ballistic currents based on the Reviewers' detailed explanations and improved the manuscript.

Note: The abstract still contains the wording "parallel to P -symmetry broken axis" which is meaningless. Can the authors explain what this means? Do they mean parallel to the mirror plane?

Thank you for your comment. As pointed out by the Reviewer, the meaningless description in the abstract is still remained. We would like to describe that it is parallel to the mirror plane. According to the Reviewer suggestion, we have revised the abstract.

Reply to the comments of Reviewer #2

Authors significantly improved their manuscript with detailed explanation of their work. I would recommend this version of the manuscript to be published in Nat. Commun.

We would like to appreciate the Reviewer's positive comment to improve our manuscript. We are also strongly encouraged by the Reviewer's comment.

Reply to the comments of Reviewer #3

After reviewing the authors' responses, it is evident that they have made some improvements to the manuscript. However, I'm not fully convinced regarding the physical mechanisms presented, and I hope the authors can think more about the below questions and provide more convincing answer or make some adjustments in the paper.

Thank you for your important and helpful comments. According to the Reviewer's advices, an additional experiment and a more in-depth discussion has provided for the revision of our manuscript.

1. The physics and technology behind the first two figures are essentially identical to the previous WSe₂/BP work. The authors should emphasize the unique contributions of their paper rather than dedicating half of the contents to what has already been shown in previous works.

Thank you for your very important comments. As pointed by the Reviewer, we admit that the first two figures are essentially similar to those in the previous study of WSe₂/BP in terms of shift current due to broken *P*-symmetry at artificial hetero-interface [T. Akamatsu, *et al.*, *Science* **372**, 6537 (2021)]. However, we think that the solid demonstration of the shift current at MoS₂/CrPS₄ artificial vdW hetero-interface due to broken *P*-symmetry has important meanings based on the various experimental results such as excitation power dependence of *I-V* curves, spontaneous photocurrent map, and so on, because the shift current in these materials combination has been reported for the first time. We also show not only the effect of shift current, but also injection current in Fig. S7 in the manuscript, which has not been reported in the previous WSe₂/BP hetero-interface. These would provide us the new insights into the bulk photovoltaic effect in the artificial vdW hetero-interface. Moreover, based on these results and discussion, the further demonstration and discussion of anomalous spontaneous photoresponses arising both from the shift current and magnetic injection current in the MoS₂/CrPS₄ hetero-interface are realized due to broken *P*-symmetry and *T*-symmetry system at low temperature.

2. As the authors are discussing all the phases in the system (FM, cAFM, AFM, non-magnetic), they should either clearly explain the mechanism behind each state or just reduce the scope of the topic. In system with inversion and time-reversal breaking symmetry, linear shift current, circular injection current, linear injection current and circular shift current all co-exist. The author needs to explicitly define and prove what are the mechanisms for a given temperature and magnetic field/magnetic state for the system. Also, when two spin states are time-reversal partners, the photocurrents tied to those states should also reverse. Several questions regarding on the topic:

Thank you for your helpful comments. As pointed out by the Reviewer, the many factors of magnetic states in the material and various bulk photovoltaic effects would make confusions of readers including the Reviewers. Thus, to help the understandings, we added the summary table for the relationship of linear shift current, circular injection

current, linear injection current and circular shift current, and for magnetic phases of CrPS₄, as shown in Table R1 and Fig. R1.

	Linear Shift current	Circular Shift current	Linear Injection current	Circular Injection current	Observed
Non-magnetic (> 40 K)	○ (Fig.2)	×	×	×	Only shift current
AFM	○ (Fig.3)	×	○ (Fig.3)	×	Shift current + “Magnetic” injection current
FM	○ (Fig.4)	×	×* (Fig.4)	×	Shift current + “Magnetic” injection current

Table. R1 Summary table of bulk photovoltaic effects and magnetic phase of CrPS₄ in vdW hetero-structure. *The linear injection current would remain only at hetero-interface.

Fig. R1 Magnetic phase of CrPS₄ as a function of temperature and external magnetic field [Y. Peng, *et al.*, *Adv Mater* **32**, 2106592 (2020)]. The experimental conditions at 10 and 35 K followed by the external magnetic field are shown by the dotted lines.

2.1 For Fig. S14, what are the magnetic phases when the field is sweeping up and down to around 3T? Why there is a hysteretic opening?

In Fig. S14, the magnetic phase in the region from 0 to 4 T is cAFM, and that in the higher magnetic field region above 4 T is FM. The hysteresis of spontaneous photocurrent is caused by the magnetic phase hysteresis.

2.2 For Fig. 4c, the two c-AFM states are time-reversal partners. Why are the photocurrent there identical?

Thank you for your very important question. The two c-AFM states in Fig. 4c are magnetic phases produced by the inverted up and down external magnetic fields, but unlike the FM state, they do not show perfectly inverted spin states along z-directions. The spin states of c-AFM in the top layer of CrPS₄ show that the spins are slightly up and down canted by the up and down external magnetic fields in Figure R2, with anti-ferromagnetically spin ordered between layers, which might provide the slight difference of magnetic injection currents in the up and down external magnetic fields. There might be slightly difference of spontaneous currents under the up and down external magnetic fields, however the more precise experiments with better signal to noise ratio are need to clearly clarify. Moreover, the theoretical predictions of magnetic injection currents in the c-AFM states are needed to refer. We think that these will be the position as important future studies for further understanding the novel phenomena experimentally observed here.

Fig. R2 Schematic of c-AFM spin conditions with up and down external magnetic fields [Y. Peng, *et al.*, *Adv Mater* **32**, 2001200 (2020)].

2.3 if the author could not determine the AFM domain, can they make a MoS₂-CrPS₄-MoS₂ sandwich structure and show top/bottom MoS₂ have same or different behavior?

Thank you for your insightful suggestion. According to the Reviewer's suggestion, we have fabricated a MoS₂/CrPS₄/MoS₂ sandwich structure device and measured the *I-V* properties. Figure R3(a) shows the schematic of the sandwich device structure, where the Au/Bi electrodes were fabricated on the top MoS₂/CrPS₄ and the bottom CrPS₄/MoS₂ were contacted to the gold electrodes on SiO₂/Si substrate. The MoS₂/CrPS₄/MoS₂ was dropped onto the pre-patterned gold electrodes to make the structure. We can successfully make the MoS₂/CrPS₄/MoS₂ sandwich structure device,

as shown in Fig. R3(a).

Figure R3(b) shows the I - V characteristics of sandwich device at room temperature. In the top electrodes, the finite spontaneous photocurrent was clearly observed, as similar to the results in Fig. 1(d). However, in the bottom electrodes, the I - V curve show very large resistivity and the spontaneous photocurrent was not observed, which might come from the contact problem between MoS₂ and gold electrodes.

We also think that the experiments of MoS₂/CrPS₄/MoS₂ might be one of the key study to further understand the results, which will be the target for the future research.

Fig. R3 (a) Optical image of MoS₂/CrPS₄/MoS₂ sandwich device. The inset shows schematic image of sandwich device. (b) I - V curves at the top and bottom side electrodes under the light illumination. The I - V curves are plotted from the difference of detected current from dark current.

3. For the newly added paragraph (line 166-179), I don't understand the logic of "however" and "thus" at line 168 and 171. Are the authors implying the system should have C₃ in line 168? And how do they conclude that the current is shift current by comparing them?

Thank you for your careful reading of our manuscript. The "however" in line 168 follow the logic that the experimental results clearly differ from the predicted theoretical calculation based on the assumption of C₃ symmetry system [B. I. Sturman, *et al.*, *The Photovoltaic and Photorefractive Effects in Non-Centrosymmetric Materials* (Ferroelectricity and Related Phenomena Vol 8) Ed GW Taylor, 1992)]. In other words, we reject the possibility that the spontaneous photocurrent observed in this experiment originate only from the C₃ symmetry system of MoS₂ on the surface. We also confirm that the spontaneous photocurrent originates from the hetero-interface with an artificially broken P -symmetry, since the experimental pattern is always positive and shows two-fold symmetry, "thus" we conclude that the shift current is due to P -symmetry breaking.

4. There are still several grammar and spelling errors

Thank you for your careful reading of our manuscript. According to the Reviewer's pointing out, we have again used the English language editing service provided by Springer Nature for this manuscript.

Reply to the comments of Reviewer #4

Thank you for your detail explanation of co-review system. We understand this co-review system and really appreciate your cooperation in reviewing our manuscript. We strongly hope that this review will contribute to provide novel experiences for the early carrier researchers.

Based on the helpful comments from the Reviewers, we added a more detailed discussion to our manuscript and we hope that we have now produced a better account for our work. We would like to express our appreciation to the Reviewers for their suggestions in improving our study. We will be grateful if you could offer us a opportunity to review to *Nature Communications* with the revisions made. Look forward to hearing from you soon.

Yours sincerely,
Kazunari Matsuda